# STATE-ACTION JOINT REGULARIZED IMPLICIT POLICY FOR OFFLINE REINFORCEMENT LEARNING

## ABSTRACT

Offline reinforcement learning enables learning from a fixed dataset, without further interactions with the environment. The lack of environmental interactions makes the policy training vulnerable to state-action pairs far from the training dataset and prone to missing rewarding actions. For training more effective agents, we propose a framework that supports learning a flexible and well-regularized policy, which consists of a fully implicit policy and a regularization through the state-action visitation frequency induced by the current policy and that induced by the data-collecting behavior policy. We theoretically show the equivalence between policy-matching and state-action-visitation matching, and thus the compatibility of many prior work with our framework. An effective instantiation of our framework through the GAN structure is provided, together with some techniques to explicitly smooth the state-action mapping for robust generalization beyond the static dataset. Extensive experiments and ablation study on the D4RL dataset validate our framework and the effectiveness of our algorithmic designs.

## 1    INTRODUCTION

Offline reinforcement learning (RL), also known as batch RL, aims at training agents from previously-collected fixed datasets that are typically large and heterogeneous, with a special emphasis on no interactions with the environment during the learning process (Ernst et al., 2005; Lange et al., 2012; Fujimoto et al., 2019; Kumar et al., 2019; Wu et al., 2019; Agarwal et al., 2020; Siegel et al., 2020; Wang et al., 2020). This paradigm extends the applicability of RL to where the environmental interactions are costly or even potentially dangerous, such as healthcare (Tseng et al., 2017; Gottesman et al., 2018; Nie et al., 2019), autonomous driving (Yurtsever et al., 2020), and recommendation systems (Swaminathan et al., 2017; Gilotte et al., 2018). While (online) off-policy RL algorithms, such as DDPG (Lillicrap et al., 2016), TD3 (Fujimoto et al., 2018), and SAC (Haarnoja et al., 2018a) could be directly adopted in an offline setting, their application can be unsuccessful (Fujimoto et al., 2019; Kumar et al., 2019), especially in high-dimensional continuous control tasks, where function approximations are inevitable and data samples are non-exhaustive. Such failures may be attributed to the shift between the state-action visitation frequency induced by the current policy and that by the data-collecting behavior policy, where unseen state-action pairs are presented to the action-value estimator, resulting in possibly uncontrollable extrapolation errors (Fujimoto et al., 2019; Kumar et al., 2019). In this regard, one approach to offline RL is to control the difference between the observed and policy-induced state-action visitation frequencies, so that the current policy mostly generates state-action pairs that have reliable action-value estimate.

Previous work in this line of research typically **(1)** regularizes the current policy to be close to the behavior policy during the training process, *i.e.*, policy or state-conditional action distribution matching; **(2)** uses a Gaussian policy class with a learnable mean and diagonal covariance matrix (Kumar et al., 2019; Wu et al., 2019). See Appendix A for a detailed review. However, at any given state $s$, the underlying action-value function over the action space may possess multiple local maxima. A deterministic or uni-modal stochastic policy may only capture one of the local optima and neglect lots of rewarding actions. An even worse situation occurs when such policies exhibit a strong mode-covering behavior, artificially inflating the density around the average of multiple rewarding actions that itself may be inferior.

Previous work under the policy-matching theme mainly takes two approaches. One approach directly estimates the divergence between the state-conditional distributions over actions (Wu et al., 2019). However, on tasks with continuous state space, with probability one, no state can appear in the dataset more than once. In other words, for each observed state $s_i$, the offline dataset has only one corresponding action $a_i$ from the behavior policy. Thus, one is only able to use a single point to assess whether the current policy is close to the data-collecting behavior policy at any particular state, which may not well reflect the true divergence between the two conditional distributions. The other approach, *e.g.*, Kumar et al. (2019), resorts to a two-step strategy: First, fit a generative model $\pi(a \,|\, s)$ to clone the behavior policy; Second, estimate the distance between the fitted behavior policy $\pi(a \,|\, s)$ and the current policy, and minimize that distance as a way to regularize. While this approach, which can acquire multiple samples from the cloned behavior policy, is able to accurately estimate the distance between the current policy and cloned behavior, its success relies heavily on how well the inferred behavior-cloning generative model mimics the true behavior policy. On tasks with large or continuous state space, however, the same problem arises that each state-conditional action distribution is fitted on only one data point. Moreover, some prior work use conditional VAE (CVAE, Sohn et al. (2015)) as the generative model to clone the possibly-multimodal behavior policy, which further suffers to the problem that CVAE may exhibit a strong mode-covering behavior that allocates large probability density to low data-density regions. Inasmuch as these weaknesses, one may naturally question on how good the samples from such a cloned policy resemble the truth, and further, on the quality of the constraint imposed by sample-based calculation of the distance between such a cloned behavior policy and the current policy.

To address these concerns, we are motivated to develop a new framework that not only supports an expressive policy, which can be as flexible as needed, but also well regularizes this flexible policy towards the data-collecting behavior policy. Specifically, **(1)** instead of using the classical deterministic or uni-modal Gaussian policy, we train a fully implicit policy for its flexibility to capture multiple modes in the action-value function; **(2)** to avoid the aforementioned potential problems in the policy-matching regularization, we directly control the distance between the state-action visitation frequency induced by the current policy and that induced by the behavior policy, as an auxiliary training target. Hence, our approach does not need to build a generative model to clone the behavior policy. On the theoretical side, we prove in Section 3 that the approach of matching the behavior and current policies is equivalent to matching their corresponding state-action visitation frequencies, which reveals the compatibility of many prior work with our framework. Similar notion of matching the state-action visitations in offline RL is taken by the DICE family (Nachum et al., 2019; Lee et al., 2021a), but they either use a Gaussian policy or a mixture of Gaussian policy with a per-dataset tuned number of mixtures. Besides, these algorithms have high computational complexity, which, together with inflexible policies and intensive hyperparameter tuning, limit their practical applicability. We instantiate our framework with a GAN structure that approximately minimizes the Jenson–Shannon divergence between the visitation frequencies. Furthermore, we design techniques to explicitly encourage robust behavior of our policy at states not covered in the static dataset. We conduct ablation study on several components of our algorithm and analyze their contributions. With all these considerations, our full algorithm achieves state-of-the-art performance on various tasks from the D4RL dataset (Fu et al., 2021), validating the effectiveness of our framework and implementation.

## 2 BACKGROUND AND MOTIVATION

We first present some background information and then introduce a toy example to illustrate the motivations of the proposed framework for offline RL.

**Offline RL.** Following the classic RL setting (Sutton & Barto, 2018), the interaction between the agent and environment is modeled as a Markov decision process (MDP), specified by the tuple $\mathcal{M} = (\mathbb{S}, \mathbb{A}, \mathcal{P}, r, \gamma)$, where $\mathbb{S}$ denotes the state space, $\mathbb{A}$ the action space, $\gamma \in (0, 1]$ the discount factor, $\mathcal{P}(s' \,|\, s, a) : \mathbb{S} \times \mathbb{S} \times \mathbb{A} \to [0, 1]$ the environmental dynamics, and $r(s, a) : \mathbb{S} \times \mathbb{A} \to [R_{\min}, R_{\max}]$ the reward function. The goal of RL is to learn a policy $\pi_\phi(a_t \,|\, s_t)$, parametrized by $\phi$, that maximizes the expected cumulative discounted reward $R_t \triangleq R(s_t, a_t) = \mathbb{E}\left[\sum_{k=0}^{\infty} \gamma^k r_{t+k+1}\right]$.

In offline RL (Fujimoto et al., 2019; Kumar et al., 2019; 2020; Levine et al., 2020), the agent only has access to a fixed dataset $\mathbb{D} \triangleq \{(s, a, r, s')\}$, consisting of transition tuples from rollouts by some behavior policies $\pi_b(a \,|\, s)$. We denote the state-action visitation frequency induced by the behavior

policy $\pi_b$ as $d_b(\boldsymbol{s}, \boldsymbol{a})$ and its state-marginal, the state visitation frequency, as $d_b(\boldsymbol{s})$. Similarly, $d_\phi(\boldsymbol{s}, \boldsymbol{a})$ and $d_\phi(\boldsymbol{s})$ are the counterparts for the current policy $\pi_\phi$. Here, $d_b(\boldsymbol{s}, \boldsymbol{a}) = d_b(\boldsymbol{s})\pi_b(\boldsymbol{a} \,|\, \boldsymbol{s})$ and we assume $\mathbb{D} \sim d_b(\boldsymbol{s}, \boldsymbol{a})$. The visitation frequencies in the dataset are denoted as $d_{\mathbb{D}}(\boldsymbol{s}, \boldsymbol{a})$ and $d_{\mathbb{D}}(\boldsymbol{s})$, which are discrete approximations to $d_b(\boldsymbol{s}, \boldsymbol{a})$ and $d_b(\boldsymbol{s})$, respectively.

**Actor-Critic Algorithm.** Denote $Q^\pi(\boldsymbol{s}, \boldsymbol{a}) = \mathbb{E}[\sum_{t=0}^\infty \gamma^t r(\boldsymbol{s}_t, \boldsymbol{a}_t) \,|\, \boldsymbol{s}_0 = \boldsymbol{s}, \boldsymbol{a}_0 = \boldsymbol{a}]$ as the action-value function. In the actor-critic scheme (Sutton & Barto, 2018), the critic $Q^\pi(\boldsymbol{s}, \boldsymbol{a})$ is often approximated by a neural network $Q_{\boldsymbol{\theta}}(\boldsymbol{s}, \boldsymbol{a})$, parametrized by $\boldsymbol{\theta}$ and trained by applying the Bellman operator (Lillicrap et al., 2016; Haarnoja et al., 2018a; Fujimoto et al., 2019) as

$$\arg\min_{\boldsymbol{\theta}} \left[ Q_{\boldsymbol{\theta}}(\boldsymbol{s}, \boldsymbol{a}) - \left( r(\boldsymbol{s}, \boldsymbol{a}) + \gamma \mathbb{E}_{\boldsymbol{s}' \sim \mathcal{P}(\cdot \,|\, \boldsymbol{s}, \boldsymbol{a}), \, \boldsymbol{a}' \sim \pi(\cdot \,|\, \boldsymbol{s}')} \left[ Q_{\boldsymbol{\theta}}(\boldsymbol{s}', \boldsymbol{a}') \right] \right) \right]^2 .$$

The actor $\pi_\phi$ aims at maximizing the expected value of $Q_{\boldsymbol{\theta}}$, with the training objective expressed as

$$\arg\max_{\boldsymbol{\phi}} \left\{ J(\pi_{\boldsymbol{\phi}}) = \mathbb{E}_{\boldsymbol{s} \sim d_{\boldsymbol{\phi}}(\boldsymbol{s}), \, \boldsymbol{a} \sim \pi_{\boldsymbol{\phi}}(\boldsymbol{a} \,|\, \boldsymbol{s})} \left[ Q_{\boldsymbol{\theta}}(\boldsymbol{s}, \boldsymbol{a}) \right] \right\}, \tag{1}$$

where $d_\phi(\boldsymbol{s})$ is the state visitation frequency under policy $\pi_\phi$. In offline RL, sampling from $d_\phi(\boldsymbol{s})$ is infeasible as no interactions with the environment are allowed. A common and practically effective approximation (Fu et al., 2019; Levine et al., 2020) to Equation 1 is

$$J(\pi_{\boldsymbol{\phi}}) \approx \mathbb{E}_{\boldsymbol{s} \sim d_b(\boldsymbol{s}), \, \boldsymbol{a} \sim \pi_{\boldsymbol{\phi}}(\boldsymbol{a} \,|\, \boldsymbol{s})} \left[ Q_{\boldsymbol{\theta}}(\boldsymbol{s}, \boldsymbol{a}) \right], \tag{2}$$

where sampling from $d_b(\boldsymbol{s})$ can be implemented easily as sampling from the offline dataset $\mathbb{D}$.

**Generative Adversarial Nets.** GAN (Goodfellow et al., 2014) provides a framework to train deep generative models, with two neural networks jointly trained in an adversarial manner: a generator $G_\phi$, parametrized by $\phi$, that fits the data distribution and a discriminator $D_{\boldsymbol{w}}$, parametrized by $\boldsymbol{w}$, that outputs the probability of a sample coming from the training data rather than $G_\phi$. Samples $\boldsymbol{x}$'s from the generator's distribution $d_\phi(\boldsymbol{x})$ are drawn via $\boldsymbol{z} \sim p_{\boldsymbol{z}}(\boldsymbol{z}), \boldsymbol{x} = G_\phi(\boldsymbol{z})$, where $p_{\boldsymbol{z}}(\boldsymbol{z})$ is some noise distribution. Both $G_\phi$ and $D_{\boldsymbol{w}}$ are trained via a two-player min-max game as

$$\min_{\boldsymbol{\phi}} \max_{\boldsymbol{w}} \left\{ V(D_{\boldsymbol{w}}, G_{\boldsymbol{\phi}}) = \mathbb{E}_{\boldsymbol{y} \sim d_{\mathbb{D}}(\cdot)} \left[ \log D_{\boldsymbol{w}}(\boldsymbol{y}) \right] + \mathbb{E}_{\boldsymbol{z} \sim p_{\boldsymbol{z}}(\boldsymbol{z})} \left[ \log(1 - D_{\boldsymbol{w}}(G_{\boldsymbol{\phi}}(\boldsymbol{z}))) \right] \right\}, \tag{3}$$

where $d_{\mathbb{D}}(\cdot)$ is the data distribution. Given the optimal discriminator $D_G^*$ at $G_\phi$, the training objective of $G_\phi$ is determined by the Jensen–Shannon divergence (JSD) (Lin, 1991) between $d_{\mathbb{D}}$ and $d_\phi$ as $V(D_G^*, G_\phi) = -\log 4 + 2 \cdot \mathrm{JSD}(d_{\mathbb{D}} \| d_\phi)$, with the global minimum achieved if and only if $d_\phi = d_{\mathbb{D}}$. Therefore, one may view GAN as a distributional matching framework that approximately minimizes the Jensen–Shannon divergence between the generator distribution and data distribution.

**Motivations.** To illustrate our motivations of training an expressive policy under an appropriate regularization, we conduct a toy experiment of behavior cloning, as shown in Figure 1, where we use the $x$- and $y$-axis values to represent the state and action, respectively. Figure 1a illustrates the state-action joint distribution of the data-collecting behavior policy that we try to mimic. For Figures 1b-1e, we use the same test-time state marginal distribution, which consists of an equal mixture of the behavior policy's state distribution and a uniform state distribution between $-1.5$ and $1.5$. If the inferred policy well approaches the behavior policy, we expect **(1)** clear concentration on the eight centers and **(2)** smooth interpolation between centers, which implies a good and smooth fit to the behavior policy. We start the toy experiment with fitting a CVAE model, a representative behavior-cloning method, to the dataset. As shown in Figure 1b, CVAE exhibits a mode-covering behavior that covers the data density modes at the expense of overestimating unwanted low data density regions. Hence, the regularization ability is questionable of using CVAE as a proxy for the behavior policy in some prior work. Replacing CVAE with the conditional GAN (CGAN, Mirza & Osindero (2014)), *i.e.*, replacing the KL loss with JSD loss, but adopting the Gaussian policy popular in prior offline RL work partially alleviates the issues but drops necessary modes, as shown in Figure 1c. This shows the inflexibility of Gaussian policies. Replacing the Gaussian policy in CGAN with an implicit policy, while still training CGAN via sampled-based policy-matching, improves the capability of capturing multiple modes, as shown in Figure 1d. Nevertheless, it is still prone to mode collapse and interpolates less smoothly between the seen states. Finally, training the implicit-policy CGAN via direct state-action-visitation joint matching leads to the best performance. As shown in Figure 1e, it concentrates clearly on the eight centers and interpolates smoothly between the seen states. Thus, constraining the state-action visitations can be an effective way to regularize implicit policies in offline RL. These observations motivate us to train implicit policies in offline RL, with sample-based regularization on the state-action visitations.

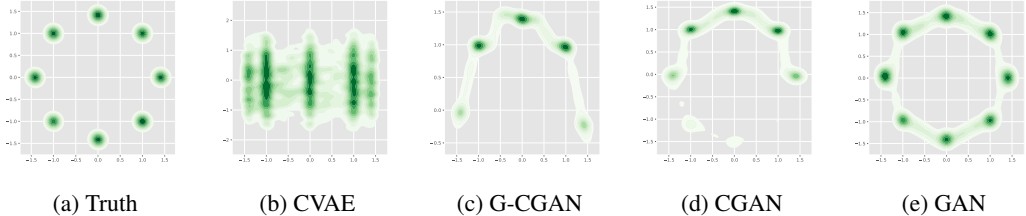

|           |           |             |           |           |
|-----------|-----------|-------------|-----------|-----------|
| (a) Truth | (b) CVAE  | (c) G-CGAN  | (d) CGAN  | (e) GAN   |

Figure 1: Performance of approximating the behavior policy on the eight-Gaussian dataset. A conditional VAE ("CVAE"), a conditional GAN ("CGAN"), and a Gaussian-generator conditional GAN ("G-CGAN") are fitted using the conditional-distribution (policy) matching approach. A conditional GAN ("GAN") is fitted using the basic state-action-joint-visitation matching strategy (Section 4.1). More details are provided in Appendix D.1.

## 3 THEORETICAL ANALYSIS

As discussed in Section 1 and detailed in Appendix A, one common theme in prior work in offline RL is controlling the distance between the behavior policy and current policy during the training process. In this section, we prove that this approach, in essence, controls the corresponding state-action visitations. This analysis theoretically links a line of prior offline-RL work with our proposed framework, manifesting the generality of our algorithmic idea. Note that $d_\phi(s, a) = d_\phi(s)\pi_\phi(a \mid s)$ and similarly for $d_b(s, a)$. Hence, it is sufficient to show the closeness between $d_\phi(s)$ and $d_b(s)$ when $\pi_\phi(a \mid s)$ is close to $\pi_b(a \mid s)$. Below we give our analysis for the matrix (finite state space) case. Continuous state-space cases may be analyzed similarly and are left for future work. The Proofs are deferred to Appendix C.

**Notation.** Denote $\boldsymbol{A}_{-i*}$ as matrix $\boldsymbol{A}$ with its $i$-th row removed; $\kappa(\boldsymbol{A})$ as the 2-norm condition number of $\boldsymbol{A}$; $\boldsymbol{1}$ as a row vector of all ones and $\boldsymbol{I}$ as an identity matrix, both with an appropriate dimension. Assume that the state space $\mathbb{S}$ is finite with cardinality $N$, i.e., $\mathbb{S} = (s^1, \ldots, s^N)$. The transition probabilities associated with policy $\pi_\phi$ over $\mathbb{S}$ is then an $N \times N$ matrix $\boldsymbol{T}_\phi$, whose $(i, j)$ entry is $\boldsymbol{T}_{\phi,(i,j)} = p_{\pi_\phi}\left(S_{t+1} = s^j \mid S_t = s^i\right) = \int_\mathbb{A} \mathcal{P}\left(S_{t+1} = s^j \mid S_t = s^i, A_t = a_t\right) \pi_\phi\left(a_t \mid s^i\right) da_t$, and similarly for $\boldsymbol{T}_b$, the transition matrix associated with $\pi_b$. Note that in this case, $d_\phi(s), d_b(s)$ are vectors and we denote $\boldsymbol{d}_\phi \triangleq \boldsymbol{d}_\phi(s), \boldsymbol{d}_b \triangleq \boldsymbol{d}_b(s) \in \mathbb{R}^N$ and $\boldsymbol{d}_\phi = \boldsymbol{d}_b + \Delta \boldsymbol{d}$.

**Theorem 1.** *Denote*

$$\kappa_{max} = \max_{i=2,\ldots,N+1} \kappa\left(\begin{pmatrix} \boldsymbol{1} \\ \boldsymbol{I} - \boldsymbol{T}_b^\top \end{pmatrix}_{-i*}\right).$$

*If*

$$\max_{i=1,\ldots,N} \left\|\pi_b\left(\cdot \mid s^i\right) - \pi_\phi\left(\cdot \mid s^i\right)\right\|_1 \le \epsilon < \frac{1}{\kappa_{max}}$$

*and $\boldsymbol{T}_{i,j}^\phi, \boldsymbol{T}_{i,j}^b > 0, \forall i, j \in \{1, 2, \ldots, N\}$, then*

$$2\text{TV}(\boldsymbol{d}_\phi, \boldsymbol{d}_b) = \|\Delta \boldsymbol{d}\|_1 = \|\boldsymbol{d}_\phi - \boldsymbol{d}_b\|_1 \le \frac{\epsilon \, \kappa_{max}}{1 - \epsilon \, \kappa_{max}} \to 0 \quad as \; \epsilon \to 0.$$

*Remark.* **(1)** We note that $\kappa_{max}$ is a constant for fixed $\boldsymbol{T}_b$ and can be calculated by iteratively removing columns of $\boldsymbol{T}_b$ and computing the SVD of the referred matrix. **(2)** The assumption that $\boldsymbol{T}_{i,j}^\phi, \boldsymbol{T}_{i,j}^b > 0, \forall i, j \in \{1, \ldots, N\}$ can be satisfied by substituting the zero entries in the original transition matrix with a small number and re-normalized each row of the resulting matrix, as in the PageRank algorithm (Page et al., 1998; Langville & Meyer, 2004).

In practice, the offline dataset $\mathbb{D}$ often consists of samples collected by several policies. Equivalently, the behavior policy $\pi_b(\cdot \mid s)$ is a mixture of single policies. Assume that there are $K$ such policies $\{\pi_{b_k}(\cdot \mid s)\}_{k=1}^K$ with mixture probabilities $\{w_k\}_{k=1}^K$, i.e., $\pi_b(\cdot \mid s) = \sum_{k=1}^K w_k \pi_{b_k}(\cdot \mid s), \sum_{k=1}^K w_k = 1$. Since we collect $\mathbb{D}$ by running each $\pi_{b_k}(\cdot \mid s)$ a proportion of $w_k$ of total time, we may decompose $\mathbb{D}$ as $\mathbb{D} = \bigcup_{k=1}^K \mathbb{D}_k$, where $\mathbb{D}_k$ consists of $w_k$ proportion of data in $\mathbb{D}$. Thus, $\boldsymbol{d}_\mathbb{D}(s) = \sum_{k=1}^K w_k \boldsymbol{d}_{\mathbb{D}_k}(s)$ and the approximation $\boldsymbol{d}_\phi(s) \approx \boldsymbol{d}_\mathbb{D}(s)$ has population version $\boldsymbol{d}_\phi(s) \approx \sum_{k=1}^K w_k \boldsymbol{d}_{b_k}(s) \triangleq \boldsymbol{d}_b(s)$. As before, denote $\boldsymbol{d}_{b_k}(s) \in \mathbb{R}^N$ as the limiting state-occupancy measure induced by $\pi_{b_k}$ on $\mathcal{M}$; $\boldsymbol{T}_{b_k}$ as the transition matrix induced by $\pi_{b_k}$ over $\mathbb{S}$; and $\boldsymbol{d}_\phi = \boldsymbol{d}_{b_k} + \Delta \boldsymbol{d}_k$. We extend Theorem 1 into this mixture of policies case as follows.

**Theorem 2.** *Denote*

$$\kappa_{max,k} = \max_{i=2,\dots,N+1} \kappa\left(\left(\begin{matrix} \mathbf{1} \\ \boldsymbol{I} - \boldsymbol{T}_{b_k}^\top \end{matrix}\right)_{-i*}\right), \quad \kappa_{max} = \max_{k=1\dots,K} \kappa_{max,k}.$$

*If*

$$\max_{k=1\dots,K} \max_{i=1,\dots,N} \left\| \pi_{b_k}\left(\cdot \,|\, \boldsymbol{s}^i\right) - \pi_{\boldsymbol{\phi}}\left(\cdot \,|\, \boldsymbol{s}^i\right) \right\|_1 \le \epsilon < \frac{1}{\kappa_{max}}$$

*and* $\boldsymbol{T}_{i,j}^{\boldsymbol{\phi}}, \boldsymbol{T}_{i,j}^{b_k} > 0, \forall\, i,j \in \{1,\dots,N\}, k \in \{1,\dots,K\}$, *then*

$$2\mathrm{TV}(\boldsymbol{d}_{\boldsymbol{\phi}}, \boldsymbol{d}_b) = \|\boldsymbol{d}_{\boldsymbol{\phi}} - \boldsymbol{d}_b\|_1 \le \sum_{k=1}^{K} w_k \frac{\epsilon\,\kappa_{max,k}}{1-\epsilon\,\kappa_{max,k}} \le \frac{\epsilon\,\kappa_{max}}{1-\epsilon\,\kappa_{max}} \to 0 \quad as\ \epsilon \to 0.$$

*In particular, if* $w_k = 1/K, \forall\, k \in \{1,\dots,K\}$, *then*

$$2\mathrm{TV}(\boldsymbol{d}_{\boldsymbol{\phi}}, \boldsymbol{d}_b) = \|\boldsymbol{d}_{\boldsymbol{\phi}} - \boldsymbol{d}_b\|_1 \le \frac{1}{K} \sum_{k=1}^{K} \frac{\epsilon\,\kappa_{max,k}}{1-\epsilon\,\kappa_{max,k}} \to 0 \quad as\ \epsilon \to 0.$$

*Remark.* We note that $\kappa_{max,k}$ is a constant for fixed $\boldsymbol{T}_{b_k}$ and $\kappa_{max}$ is a constant for fixed $\{\pi_{b_k}\}_{k=1}^{K}$.

We notice that similar analysis has been given in the prior work of bounding $D_{\mathrm{KL}}\left(d_{\boldsymbol{\phi}}(\boldsymbol{s}) \| d_b(\boldsymbol{s})\right)$ by $O\left(\epsilon/(1-\gamma)^2\right)$ (Schulman et al., 2015; Levine et al., 2020). However, this prior work deals with (unnormalized) discounted visitation frequencies while our bound is devoted to undiscounted visitation frequencies, since neither the data collection (*i.e.*, policy rollout) nor the state-action-visitation regularization involve the discount factor. In short, the definitions of $d_{\boldsymbol{\phi}}(\boldsymbol{s})$ and $d_b(\boldsymbol{s})$ in our work are different from this prior work. Note that this prior work depends on $1 - \gamma$ on the denominator and hence cannot be applied to the undiscounted case (discount factor $\gamma = 1$). Furthermore, instead of the KL divergence, we bound the total variation distance between the state visitation frequencies, which is a well-defined metric.

## 4 STATE-ACTION JOINT REGULARIZED IMPLICIT POLICY

In this section we discuss an instance of our framework that will be used in our empirical study in Section 5. Concretely, we train a fully implicit policy via a GAN structure to approximately minimizes the JSD between the state-action visitation frequency induced by the current policy and that induced by the behavior policy. Our basic algorithm is discussed in Section 4.1, followed by three enhancing components presented in Section 4.2 to build up our full algorithm.

This instantiation manifests three facets we consider important in offline RL: **(1)** the flexibility of the policy class, **(2)** an effective sample-based regularization without explicitly modelling the behavior policy, and **(3)** smoothness of the learned policy.

### 4.1 BASIC ALGORITHM

Motivated by the standard actor-critic and GAN frameworks described in Section 2, our basic algorithm consists of a critic $Q_{\boldsymbol{\theta}}$, an actor $\pi_{\boldsymbol{\phi}}$, and a discriminator $D_{\boldsymbol{w}}$. For training stability, we adopt the double Q-learning (Hasselt, 2010) and target network formulation to train a pair of critics $Q_{\boldsymbol{\theta}_1}, Q_{\boldsymbol{\theta}_2}$ and the target networks $Q_{\boldsymbol{\theta}_1'}, Q_{\boldsymbol{\theta}_2'}, \pi_{\boldsymbol{\phi}'}$.

To penalize uncertainty in the action-value estimates at future states, while controlling conservatism, we follow Fujimoto et al. (2019) and Kumar et al. (2019) to use the following critic-training target:

$$\widetilde{Q}\left(\boldsymbol{s}, \boldsymbol{a}\right) \triangleq r(\boldsymbol{s}, \boldsymbol{a}) + \gamma \mathbb{E}_{\boldsymbol{a}' \sim \pi_{\boldsymbol{\phi}'}(\cdot \,|\, \boldsymbol{s}')} \left[ \lambda \min_{j=1,2} Q_{\boldsymbol{\theta}_j'}\left(\boldsymbol{s}', \boldsymbol{a}'\right) + (1-\lambda) \max_{j=1,2} Q_{\boldsymbol{\theta}_j'}\left(\boldsymbol{s}', \boldsymbol{a}'\right) \right] \quad (4)$$

with some hyperparameter $\lambda \in [0,1]$, where one $\boldsymbol{a}'$ is sampled at each $\boldsymbol{s}'$ in the mini-batch. The training objective for both critic networks is minimizing the mean-squared-error between their respective action-value estimates $Q_{\boldsymbol{\theta}_j}\left(\boldsymbol{s}, \boldsymbol{a}\right)$ and $\widetilde{Q}\left(\boldsymbol{s}, \boldsymbol{a}\right)$ over state-action pairs in the mini-batch.

Our actor consists of three components: an implicit policy, state-action-visitation joint regularization, and a conservative training target.

**Implicit Policy.** As discussed in Sections 1 and 2, a deterministic or Gaussian policy may miss important rewarding actions, or even concentrate on inferior average actions. For online off-policy

RL, Yue et al. (2020) shows the benefit of introducing an implicit distribution mixed Gaussian policy. Generalizing that idea to offline RL, we train a fully implicit policy, which transforms a given noise distribution into the state-conditional action distribution via a neural network, in reminiscent of the generator in CGAN. Specifically, given $s$,

$$a \sim \pi_\phi(\cdot \mid s) = \pi_\phi(s, z), \quad z \overset{iid}{\sim} p_z(z) \tag{5}$$

where $\pi_\phi$ is a deterministic function and $p_z(z)$ is some noise distribution. As shown in Figures 1d and 1e, an implicit policy has stronger capability of learning a multi-modal policy, if needed.

**State-action Joint Regularization.** As discussed in Section 2, directly sampling from $d_\phi(s)$ is infeasible in offline RL. Motivated by Equation 2, we approximate $d_\phi(s)$ by $d_b(s)$ in our implementation, which allows an easy sampling scheme that simply samples $s$ from $\mathbb{D}$, without needing an importance sampling correction that may possess high variance (Liu et al., 2018). Such an approximation is classical in the off-policy and offline RL literature (Degris et al., 2012; Levine et al., 2020), with demonstrated empirical effectiveness in offline RL (Fu et al., 2019), besides its usage in off-policy RL (Silver et al., 2014; Schulman et al., 2015; Lillicrap et al., 2016). With this simplification , we can efficiently maximize the similarity between $d_b(s, a)$ and $d_\phi(s, a)$ with respect to sample-based estimate of some statistical divergence, such as the Jensen–Shannon divergence. Using notations in GAN, the generator sample $x$ and the data sample $y$ are defined as

$$x \triangleq (\tilde{s}, \tilde{a}), \; \tilde{s} \sim \mathbb{D}, \; \tilde{a} \sim \pi_\phi(\cdot \mid \tilde{s}); \; y \triangleq (s, a) \sim \mathbb{D}, \; \tilde{s} \text{ independent of } s \tag{6}$$

We then constraint this statistical divergence value, named generator loss $\mathcal{L}_g(\phi)$, in the training of actor. In our instantiation of approximately minimizing JSD via GAN, $\mathcal{L}_g = \mathbb{E}_x \left[ \log \left( 1 - D_w(x) \right) \right]$.

Our choice of directly matching state-action joint visitations mitigates the issue of uncontrollable extrapolation errors in the action-value function estimate, since proximity of visitation frequencies leads to reduced chances of estimating the action-values of state-action pairs far from the offline dataset. Furthermore, the problem in the policy-matching approach of fitting each state-conditional action distributions on only one data point can be circumvented, since state-action pairs $(s_i, a_i)$ in the offline dataset are all viewed as samples from the joint visitation frequency, instead of each pair being separately viewed as one sample from the state-conditional distribution, *i.e.*, $a_i \sim \pi \left( \cdot \mid s_i \right)$. Besides, the state-action-visitation joint matching approach implicitly encourages the smoothness of the state-action mapping, namely, similar states should have similar actions. This is because, for example, the discriminator in GAN can easily discriminate as "fake" a generator sample $x$ should it has state similar to a data sample but action very different from. This smoothness feature helps ensure a reliable generalization of our policy to unseen states.

**Actor-Training Target.** To prevent the accumulation of accidental errors in the training process and in the approximation of action-value function, we adopt the strategy in Kumar et al. (2019) to train the policy with respect to a conservative estimate of the action-values. Specifically, we exploit the double-Q structure and use the minimum of the two action-value estimates for policy improvement. For the ease of optimization, we use the Lagrange form of the constraint optimization problem and penalize the generator loss $\mathcal{L}_g(\phi)$ while improving the policy. Our policy-training target is

$$\arg\min_\phi -\mathbb{E}_{s \sim \mathbb{D}} \mathbb{E}_{a \sim \pi_\phi(\cdot \mid s)} \left[ \min_{j=1,2} Q_{\theta_j}(s, a) \right] + \alpha \cdot \mathcal{L}_g(\phi), \tag{7}$$

where $\alpha$ is a fixed Lagrange multiplier. At the test time, we follow Fujimoto et al. (2019) and Kumar et al. (2019) to first sample 10 actions from $\pi_\phi$ and then execute the action that maximizes $Q_{\theta_1}$.

The discriminator is trained to better distinguish samples from $d_\phi(s, a)$ and from $d_b(s, a)$. It aids the matching of the state-action-visitation frequencies through outputting $\mathcal{L}_g(\phi)$. As an example, for approximately minimizing JSD via GAN, the discriminator outputs the probability that the input, either the $x$ or $y$ in Equation 6, comes from $d_b(s, a)$. In this case, the discriminator is trained to minimize the error in assigning $x$ as "fake" and $y$ as "true," the inner maximization of Equation 3.

## 4.2 Enhancing Components

In this section we present three enhancing components for further improving our basic algorithm.

**Multiple Action-samples at Bellman Backup.** The expectation part of the critic learning target in Equation 4 can be better estimated, in terms of a smaller sample variance, by averaging over $N_a$ actions $a'$ at each $s'$ in the mini-batch, rather than just one $a'$ as in Section 4.1.

**State-smoothing at Bellman Backup.** Due to the stochastic nature of environmental transition, multiple next states $s'$ are possible after taking action $a$ at state $s$, while the offline dataset $\mathbb{D}$ only contains one such $s'$. Since the agent is unable to interact with environment to collect more data in offline RL, local exploration (Sinha et al., 2021) in the state-space appears as an effective strategy to regularize the Bellman backup by taking consideration of states close to the records in the offline dataset. We assume that: **(1)** a small transformation to a state results in states physically plausible in the underlying environment (as in Sinha et al. (2021)); **(2)** when the state space is continuous, the transition kernel $\mathcal{P}\left(\cdot \mid s, a\right)$ is locally continuous and centers at the recorded $s'$ in the dataset.

With these assumptions, we propose to fit $Q_{\boldsymbol{\theta}}(s, a)$ on the value of a small region around the recorded next state $s'$. Specifically, with a pre-specified standard deviation $\sigma_B$, we sample around $s'$ as $\hat{s} = s' + \epsilon, \epsilon \sim \mathcal{N}(\mathbf{0}, \sigma_B^2 \boldsymbol{I})$, and modify Equation 4 as

$$\widetilde{Q}\left(s, a\right) \triangleq r(s, a) + \gamma \mathbb{E}_{\hat{s}} \mathbb{E}_{\hat{a} \sim \pi_{\phi'}(\cdot \mid \hat{s})} \left[ \lambda \min_{j=1,2} Q_{\boldsymbol{\theta}_j'}\left(\hat{s}, \hat{a}\right) + (1 - \lambda) \max_{j=1,2} Q_{\boldsymbol{\theta}_j'}\left(\hat{s}, \hat{a}\right) \right], \quad (8)$$

where $N_B$ $\hat{s}$ are sampled to estimate the expectation. This strategy is equivalent to using a Gaussian distribution centered at $s'$ to approximate the otherwise non-smooth $\delta_{s'}$ transition kernel manifested in the offline dataset. Similar technique is also considered as the target policy smoothing regularization in Fujimoto et al. (2018), though smoothing therein is applied on the target action.

**State-smoothing at Joint-matching.** As described in Section 4.1, we approximate $d_{\phi}(s)$ by $d_{\mathbb{D}}(s)$ in the sample-based estimate of the chosen statistical distance. However, $d_{\mathbb{D}}(s)$ is in essence discrete and the idea of smoothing the discrete state-distribution can be applied again to provide a better coverage of the state space. This design explicitly encourages a predictable and smooth behavior at states unseen in the offline dataset. Specifically, with some pre-specified $\sigma_J^2$, we modify the sampling scheme of $\tilde{s}$ in Equation 6 as

$$\tilde{s} \sim \mathbb{D}, \ \epsilon \sim \mathcal{N}\left(\mathbf{0}, \sigma_J^2 \boldsymbol{I}\right), \ \tilde{s} \leftarrow \tilde{s} + \epsilon. \tag{9}$$

Our strategy is akin to sampling from a kernel density approximation (Wasserman, 2006) of $d_{\phi}(s)$ with data points $s \in \mathbb{D}$ and with radial basis kernel of bandwidth $\sigma_J$.

Algorithm 1 shows the main steps of our algorithm, instantiated by approximately minimizing JSD via GAN, and a detailed version is in Appendix B. Implementation details are in Appendix D.2.

---

**Algorithm 1** State-Action Joint Regularized Implicit Policy (GAN-JSD, Main Steps)

---

Initialize policy network $\pi_{\phi}$, critic network $Q_{\boldsymbol{\theta}_1}$ and $Q_{\boldsymbol{\theta}_2}$, discriminator network $D_{\boldsymbol{w}}$.
**for** each iteration **do**
 Sample a mini-batch of transitions $\mathcal{B} = \{(s, a, r, s')\} \sim \mathbb{D}$.
 From Equation 8, train the critics by $\arg\min_{\boldsymbol{\theta}_j}(Q_{\boldsymbol{\theta}_j}\left(s, a\right) - \widetilde{Q}(s, a))^2$ over $(s, a) \in \mathcal{B}$, for $j = 1, 2$.
 Calculate generator loss $\mathcal{L}_g$ using $D_{\boldsymbol{w}}$ and the $x$, $y$ in Equation 6, with state-smoothing in Section 4.2.
 Optimize policy network $\pi_{\phi}$ by Equation 7.
 Optimize discriminator $D_{\boldsymbol{w}}$ to maximize $\mathbb{E}_{\boldsymbol{y} \sim d_{\mathbb{D}}(\cdot)}\left[\log D_{\boldsymbol{w}}\left(\boldsymbol{y}\right)\right] + \mathbb{E}_{\boldsymbol{x}}\left[\log\left(1 - D_{\boldsymbol{w}}(\boldsymbol{x})\right)\right]$.
**end for**

---

## 5 EXPERIMENTS

As discussed in Section 1, we consider it important the flexibility of the policy class, the well-regularization and the smoothness of the learned policy. To this end, we develop our state-action joint regularized implicit policy, which is a framework that supports training an expressive implicit policy via an effective sample-based regularization without an explicit modelling of the behavior policy, and with techniques encouraging its smoothness. In this section we will test an instantiation of our framework on the continuous-control RL tasks. Specifically, we will first show the effectiveness of implicit policy, state-action-visitation matching, and our full algorithm (Section 5.1). We then show in ablation study (Section 5.2) the contributions of several building blocks.

**Implementation.** We use GAN to approximately control the Jensen–Shannon divergence between the state-action visitation frequencies. Our source code builds on the official BCQ repository and largely follows its network architectures. Our implementation of the GAN structure and the hyper-parameter choices follow the literature (Goodfellow et al., 2014; Mirza & Osindero, 2014; Radford

et al., 2016; Salimans et al., 2016; White, 2016; Goodfellow, 2017). To mimic a hyperparameter-agnostic setting, we minimize hyperparameter tuning across datasets. Implementation details and hyperparameter setting of our full algorithm with GAN joint-matching is in Appendix D.2.1.

## 5.1 MAIN RESULTS

To validate the effectiveness of our framework, we test three implementations of the GAN instantiation: **(1)** basic algorithm (Section 4.1) regularized by the classical policy-matching[1] ("GAN-Cond:Basic"), **(2)** basic algorithm regularized by the state-action-visitation joint matching ("GAN-Joint:Basic"), **(3)** full algorithm, which adds state-smoothing techniques onto our basic algorithm ("GAN-Joint"). We compare them with the state-of-the-art (SOTA) offline-RL algorithms: BEAR (Kumar et al., 2019), BRAC (Wu et al., 2019) (BRAC-v: value penalty), BCQ (Fujimoto et al., 2019), and CQL (Kumar et al., 2020); together with offline soft actor-critic (SAC) (Haarnoja et al., 2018a). We re-run CQL using the official source code (details in Appendix D.2.2). Results for other algorithms are from Fu et al. (2021). Table 1 validates the effectiveness of our full algorithm, together with the efficacy of the implicit policy, state-action-visitation matching, and state-smoothing.

Our full algorithm on average outperforms the baseline algorithms, and its performance is relatively stable across tasks and datasets that possess diverse nature. Our full algorithm especially shows robust and comparatively-good performance on the high-dimensional Adroit tasks and the Maze2D tasks that are collected by non-Markovian policies, which may traditionally be considered as hard in offline RL. On the Gym-MuJoCo domain, our full algorithm shows its ability to learn from datasets collected by a mixture of behavior policies, and from medium-quality examples, which are likely the datasets to encounter in real-world application. These results support our design of implicit policy, matching the state-action visitation frequencies, and explicit state-smoothing.

Comparing "GAN-Cond:Basic" with the baseline algorithms, especially BEAR and CQL that use Gaussian policies, we see that an implicit policy does in general help the performance. This aligns with our intuition in Sections 1 and 2 of the incapability of the uni-modal Gaussian policy in capturing multiple action-modes.

To verify the gain of our joint-visitation-matching approach over the classical policy matching, apart from the comparison between our full algorithm with BEAR and BRAC, the SOTA policy-matching algorithms, we further compare "GAN-Joint:Basic" with "GAN-Cond:Basic." We see that on 11 out of 16 datasets, "GAN-Joint:Basic" wins against "GAN-Cond:Basic," while results on other datasets are close. This empirical result may be attributed to the advantage of state-action-visitation matching, *e.g.*, good regularization and smoothness in the state-action mapping (Section 4.1).

Comparing "GAN-Joint" with "GAN-Joint:Basic," we see that our state-smoothing techniques do in general help the performance. This performance gain may be related to a smoother action-choice at states not covered by the offline dataset, and a more regularized Bellman backup. Note that different datasets may require different smoothing strength, and thus this comparison can potentially be more significant should one is allowed to per-dataset tune the smoothing hyperparameter.

## 5.2 ABLATION STUDY

The ablation study serves to answer the following questions: **(a):** Is implicit policy better than Gaussian policy in our framework? **(b):** Does state-smoothing at joint-matching help? **(c):** Does state-smoothing at Bellman backup matter? **(d):** How important is the standard deviation of the Gaussian noise injected in state-smoothing? Unless stated otherwise, hyperparameters for all algorithmic variants in all datasets are in Table 2.

**(a):** We compare the performance of our basic algorithm with its variant where the implicit policy therein is replaced by a Gaussian policy. To make a fair comparison, the critics, discriminator, and hyperparameter setting remain the same. Technical details are in Appendix D.2.3.

Table 4 presents the results. On 11 out of 16 datasets, our basic GAN-joint-matching algorithm has higher average return than the Gaussian policy variant. This empirical result coincides with our intuition in Section 4.1 and results in Section 5.1 that a Gaussian policy is less flexible to capture all

---

[1]Same state is used in generator and data sample, *i.e.*, $\tilde{s} = s$. Implementation follows Wu et al. (2019).

Table 1: Normalized returns for experiments on the D4RL suite of tasks. We perform experiments on tasks from the Maze2D, Gym-Mojoco, and Adroit domains. High average scores and low average ranks are desirable.

| Task Name | SAC-off | BEAR | BRAC-v | BCQ | CQL | GAN-Cond:Basic | GAN-Joint:Basic | GAN-Joint |
|---|---|---|---|---|---|---|---|---|
| maze2d-umaze | 88.2 | 3.4 | -16.0 | 12.8 | 50.5 | $52.3 \pm 19.6$ | $57.6 \pm 11.0$ | $40.1 \pm 16.9$ |
| maze2d-medium | 26.1 | 29.0 | 33.8 | 8.3 | 30.7 | $42.6 \pm 18.2$ | $39.4 \pm 10.3$ | $69.6 \pm 25.6$ |
| maze2d-large | -1.9 | 4.6 | 40.6 | 6.2 | 43.7 | $36.9 \pm 17.9$ | $52.1 \pm 20.8$ | $71.3 \pm 26.0$ |
| halfcheetah-medium | -4.3 | 41.7 | 46.3 | 40.7 | 39.0 | $43.8 \pm 0.2$ | $43.7 \pm 0.5$ | $44.1 \pm 0.3$ |
| walker2d-medium | 0.9 | 59.1 | 81.1 | 53.1 | 60.2 | $65.5 \pm 6.8$ | $70.0 \pm 9.1$ | $69.3 \pm 8.6$ |
| hopper-medium | 0.8 | 52.1 | 31.1 | 54.5 | 34.5 | $67.5 \pm 21.3$ | $66.5 \pm 15.4$ | $60.1 \pm 27.3$ |
| halfcheetah-medium-replay | -2.4 | 38.6 | 47.7 | 38.2 | 43.4 | $32.3 \pm 2.4$ | $31.3 \pm 2.8$ | $33.1 \pm 2.3$ |
| walker2d-medium-replay | 1.9 | 19.2 | 0.9 | 15.0 | 16.4 | $6.9 \pm 2.3$ | $9.9 \pm 2.0$ | $10.2 \pm 2.4$ |
| hopper-medium-replay | 3.5 | 33.7 | 0.6 | 33.1 | 29.5 | $25.6 \pm 1.6$ | $36.5 \pm 5.2$ | $29.5 \pm 2.5$ |
| halfcheetah-medium-expert | 1.8 | 53.4 | 41.9 | 64.7 | 34.5 | $76.0 \pm 9.2$ | $74.2 \pm 6.1$ | $75.8 \pm 10.1$ |
| walker2d-medium-expert | -0.1 | 40.1 | 81.6 | 57.5 | 79.8 | $73.3 \pm 13.8$ | $76.5 \pm 15.1$ | $71.2 \pm 22.0$ |
| hopper-medium-expert | 1.6 | 96.3 | 0.8 | 110.9 | 103.5 | $68.0 \pm 21.3$ | $70.9 \pm 26.8$ | $99.9 \pm 29.0$ |
| pen-human | 6.3 | -1.0 | 0.6 | 68.9 | 2.1 | $52.9 \pm 15.9$ | $61.0 \pm 13.7$ | $45.5 \pm 24.5$ |
| pen-cloned | 23.5 | 26.5 | -2.5 | 44.0 | 1.5 | $19.4 \pm 14.5$ | $31.3 \pm 21.5$ | $18.0 \pm 14.4$ |
| pen-expert | 6.1 | 105.9 | -3.0 | 114.9 | 95.9 | $126.1 \pm 17.7$ | $129.1 \pm 14.4$ | $141.1 \pm 14.8$ |
| door-expert | 7.5 | 103.4 | -0.3 | 99.0 | 87.9 | $100.8 \pm 3.8$ | $104.1 \pm 3.6$ | $103.4 \pm 3.7$ |
| Average Score | 10.0 | 44.1 | 24.1 | 51.4 | 47.1 | **55.6** | **59.6** | **61.4** |
| Average Rank | 6.8 | 4.8 | 5.5 | 4.3 | 4.6 | **3.9** | **2.9** | **3.2** |

the rewarding actions, of which an implicit policy is likely to be capable. Appendix F visualizes this comparison and shows in plots that Gaussian policies do leave out action modes in offline RL.

**(b):** We compare our full algorithm with its variant of no state-smoothing in the matching of state-action visitations. The results are shown in Table 5, where our full algorithm overall performs better than the no state-smoothing variant. This performance gain may be attributed to a better coverage of the state-space by the smoothed state-distribution (Section 4.2), which is related to a more predictable and smoother action choice at unseen states. As stated previously, this comparison can potentially be more significant should one is allowed to per-dataset tune the $\sigma_J$ parameter.

**(c):** We compare our full algorithm with its variant of no state-smoothing in the Bellman backup. Table 6 shows the results. Again, overall our full algorithm performs better than the no state-smoothing version, showing the benefit of smoothing the empirical transition kernel $\delta_{s'}$ (Section 4.2), e.g., taking the stochasticity of state-transitions into account. As before, we use the same smoothing strength across all datasets, while a per-dataset tuning of the $\sigma_B$ parameter may improve the distinction.

**(d):** To simplify hyperparameter tuning, in actual implementation we fix $\sigma_B = \sigma_J \triangleq \sigma$ (see Appendix D.2.1). To test the robustness to the $\sigma$ hyperparameter, we run our full algorithm under $\sigma \in \left\{ 1 \times 10^{-2}, 3 \times 10^{-3}, 1 \times 10^{-3}, 3 \times 10^{-4}, 1 \times 10^{-4}, 0 \right\}$. Table 7 shows the normalized returns. We see that our algorithm is relatively insensitive to the setting of $\sigma$, especially in the range $\sigma \in \left[ 1 \times 10^{-4}, 1 \times 10^{-3} \right)$ where the performance varies little with $\sigma$. A too-small $\sigma$ cannot provide enough smoothing to the state distributions. On the contrary, a too-large $\sigma$ may highly distort the information contained in the offline dataset, such as the state-transition kernel. In both cases, a degradation in the overall performance is expected.

## 6    CONCLUSION AND FUTURE WORK

In this paper, we develop a framework that supports learning a flexible while well-regularized policy in offline RL. Specifically, we train a fully implicit policy via regularization on the difference between the state-action visitation frequency induced by the current policy and that induced by the data-collecting behavior policy. An effective instantiation of our framework through the GAN structure is provided for approximately minimizing the JSD between the visitation frequencies. Other divergence metrics, such as the MMD, may also be applied and are left for future work. We further augment our algorithm with explicit state-smoothing techniques to enhance its generalizability on states beyond the offline dataset. On the theoretical side, we show the equivalence between policy-matching and state-action-visitation matching, and hence the compatibility of many prior algorithms with our framework. We note that our implementation-wise simplification of approximating current policy's state visitation frequency by the behavior's can be improved by a more tactful approximation of current policy's state occupancy measure, which is left for future work. Nevertheless, the effectiveness of our framework and implementations is validated through extensive experiment and ablation study on the D4RL dataset.

## REPRODUCIBILITY STATEMENT

To help reproduce our empirical work, we provide detail algorithmic description in Appendix B and implementation details, including the hyperparameter choice and model architecture, in Appendix D. For reproducing our theoretical results, a step-by-step proof is provided in Appendix C.

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

# A   RELATED WORK

**Offline Reinforcement Learning.** Three major themes currently exist in offline-RL research. The first focuses on more robustly estimate the action-value function (Agarwal et al., 2020; Gulcehre et al., 2021) or provide a conservative estimate of the Q-values (Kumar et al., 2020; Yu et al., 2021; Sinha et al., 2021), which may better guide the policy optimization process. The second research theme aims at designing a tactful behavior-cloning scheme so as to learn only from "good" actions in the offline dataset (Wang et al., 2020; Chen et al., 2021). Most similar to our approach, the third line of research tries to constraint the current policy to be close to the behavior policy during the training process, under the notion that Q-value estimates at unfamiliar state-action pairs can be pathologically worse due to a lack of supervised training. Specifically, Kumar et al. (2019) and Wu et al. (2021) use conditional variational autoencoder (CVAE) (Kingma & Welling, 2013; Sohn et al., 2015) to train a behavior cloning policy to sample multiple actions at each state for calculating the MMD constraint. Wu et al. (2019), Siegel et al. (2020), and Cang et al. (2021) fit a (Gaussian) behavioral prior to the offline dataset trained by (weighted) maximum likelihood objective. Jaques et al. (2019) consider a pre-trained generative prior of human dialog data before applying KL-control to the current policy. Note that these work essentially constraint the distance between the current policy and the cloned behavior policy. Laroche & Trichelair (2019) assume a known stochastic data-collecting behavior policy. Fujimoto et al. (2019) and Urpí et al. (2021) implicitly control the state-conditional action distribution by decomposing action into a behavior cloning component, trained by fitting a CVAE onto the offline data, and a perturbation component, trained to optimize the (risk-averse) returns. Besides, some work, such as Wu et al. (2019), directly estimates and regularizes the divergence between the state-conditional action distributions. In the paper, we instead take on the perspective of matching the state-action joint visitation frequencies, which avoids using a single point to estimate the divergence between distribution and removes the need for a good approximator to the behavior policy, while implicitly smooths the mapping from state to actions (Section 1). Note that if we use the same states in constructing the generator samples and the data samples, *i.e.*, $\tilde{s} = s$ in Equation 6, our implementation of state-action-visitation matching is fundamentally equivalent to matching the state-conditional action distributions, as in some prior work.

**Online Off-policy Reinforcement Learning.** A large class of modern online off-policy deep reinforcement learning algorithms trains the policy using experience replay buffer (Lin, 1992), which is a storage of the rollouts of past policies encountered in the training process (Mnih et al., 2013; Lillicrap et al., 2016; Haarnoja et al., 2017; Fujimoto et al., 2018; Haarnoja et al., 2018b; Kuznetsov et al., 2020; Yue et al., 2020; Lee et al., 2021b). This approach essentially use the state-visitation frequency of past policies to approximate that of the current policy (Equation 2). We adopt this notion in both the policy improvement step and in the implementation of state-action-visitation joint matching, with additional smoothing techniques to encourage the smoothness of the empirical transition kernel and the approximated state-visitation frequency of current policy (Section 4.2).

**Computational Distribution Matching.** Many computationally efficient algorithms exist to match two probability distributions with respective to some statistical divergence. GAN (Goodfellow et al., 2014) approximately minimizes the Jensen-Shannon divergence between the the model's distribution and the data-generating distribution. Similar adversarial training strategy is further applied to estimate a class of statistical divergence, termed the integral probability metrics (Müller, 1997), in a sample-based manner. For example, Arjovsky et al. (2017); Gulrajani et al. (2017); Miyato et al. (2018) estimate the Wasserstein-1 distance by enforcing the Lipschitz norm of the witness function to be bounded by 1. Li et al. (2017); Binkowski et al. (2018) consider Maximum Mean Discrepancy (MMD) (Gretton et al., 2012) with learnable kernels. Bellemare et al. (2017) studies the energy distance, an instance of the MMD (Sejdinovic et al., 2013). Another important class of statistical divergence, the optimal transport distance, is also studied under a computational lens. Peyré & Cuturi (2019) summarizes recent developments in computational optimal transport. Cuturi (2013); Feydy et al. (2019) study sinkhorn-based divergence, which solve an entropic regularized form of the optimal transport problem (Schrödinger, 1932) and may result in a physically more feasible distance measure in some real-world application. Zheng & Zhou (2021) contributes to this line of research by developing conditional transport algorithm, which possesses a better control over the mode-covering and mode-seeking behaviors in distributional matching. In this paper, we consider the classical GAN structure to approximately control the JSD between the state-action visitation frequencies, since the GAN structure is simple, effective and well-studied. Other divergence metrics may also be applicable in our framework and are left for future work.

# B  FULL ALGORITHM

---

**Algorithm 2** State-Action Joint Regularized Implicit Policy (GAN-JSD, Detailed Version)

---

**Input:** Learning rate $\eta_{\boldsymbol{\theta}}, \eta_{\boldsymbol{\phi}}, \eta_{\boldsymbol{w}}$; target smoothing factor $\beta$; noise distribution $p_{\boldsymbol{z}}(\boldsymbol{z})$; policy network $\pi_{\boldsymbol{\phi}}$ with parameter $\boldsymbol{\phi}$; critic network $Q_{\boldsymbol{\theta}_1}$ and $Q_{\boldsymbol{\theta}_2}$ with parameters $\boldsymbol{\theta}_1, \boldsymbol{\theta}_2$; discriminator network $D_{\boldsymbol{w}}$ with parameter $\boldsymbol{w}$; generator loss $\mathcal{L}_g$; standard deviation $\sigma_B, \sigma_J$; number of smoothed states $N_B$; number of actions $\boldsymbol{a}'$ at $\boldsymbol{s}'$ $N_a$; number of epochs for warm start $N_{\mathrm{warm}}$; policy frequency $k$.
**Output:** Learned neural network parameters $\boldsymbol{\phi}, \boldsymbol{\theta}_1, \boldsymbol{\theta}_2, \boldsymbol{w}$.
{// Initialization} Initialize $\boldsymbol{\phi}, \boldsymbol{\theta}_1, \boldsymbol{\theta}_2, \boldsymbol{w}$. Initialize $\boldsymbol{\phi}' \leftarrow \boldsymbol{\phi}, \boldsymbol{\theta}'_1 \leftarrow \boldsymbol{\theta}_1, \boldsymbol{\theta}'_2 \leftarrow \boldsymbol{\theta}_2$. Load dataset $\mathcal{D}$.
**for** each epoch **do**
    **for** each iteration within current epoch **do**
        Sample a mini-batch of transitions $\mathcal{B} = \{(\boldsymbol{s}, \boldsymbol{a}, r, \boldsymbol{s}')\} \sim \mathbb{D}$.
        {// Policy Evaluation}
        For each $\boldsymbol{s}' \in \mathcal{B}$ sample $N_B$ $\hat{\boldsymbol{s}}$ with noise standard deviation $\sigma_B$ for state-smoothing via,

$$\hat{\boldsymbol{s}} = \boldsymbol{s}' + \boldsymbol{\epsilon}, \boldsymbol{\epsilon} \sim \mathcal{N}(\boldsymbol{0}, \sigma_B^2 \boldsymbol{I}).$$

        Sample $N_a$ corresponding actions $\hat{\boldsymbol{a}} \sim \pi_{\boldsymbol{\phi}'}(\cdot \mid \hat{\boldsymbol{s}})$ for each $\hat{\boldsymbol{s}}$.
        Calculate $\widetilde{Q}(\boldsymbol{s}, \boldsymbol{a})$ as

$$\widetilde{Q}(\boldsymbol{s}, \boldsymbol{a}) \triangleq r(\boldsymbol{s}, \boldsymbol{a}) + \gamma \frac{1}{N_B \times N_a} \sum_{(\hat{\boldsymbol{s}}, \hat{\boldsymbol{a}})} \left[ \lambda \min_{j=1,2} Q_{\boldsymbol{\theta}'_j}(\hat{\boldsymbol{s}}, \hat{\boldsymbol{a}}) + (1 - \lambda) \max_{j=1,2} Q_{\boldsymbol{\theta}'_j}(\hat{\boldsymbol{s}}, \hat{\boldsymbol{a}}) \right].$$

        Minimize the critic loss with respect to $\boldsymbol{\theta}_j, j = 1, 2$, over $(\boldsymbol{s}, \boldsymbol{a}) \in \mathcal{B}$, with learning rate $\eta_{\boldsymbol{\theta}}$,

$$\arg\min_{\boldsymbol{\theta}_j} \frac{1}{|\mathcal{B}|} \sum_{(\boldsymbol{s}, \boldsymbol{a}) \in \mathcal{B}} \left( Q_{\boldsymbol{\theta}_j}(\boldsymbol{s}, \boldsymbol{a}) - \widetilde{Q}(\boldsymbol{s}, \boldsymbol{a}) \right)^2.$$

        {// Policy Improvement}
        Resample $|\mathcal{B}|$ new states $\tilde{\boldsymbol{s}} \sim \mathbb{D}$ independent of $\boldsymbol{s}$. Add state-smoothing to $\tilde{\boldsymbol{s}}$ with noise standard deviation $\sigma_J$ using $\boldsymbol{\epsilon} \sim \mathcal{N}(\boldsymbol{0}, \sigma_J^2 \boldsymbol{I})$, $\tilde{\boldsymbol{s}} \leftarrow \tilde{\boldsymbol{s}} + \boldsymbol{\epsilon}$.
        Form the generator sample $\boldsymbol{x}$ and data sample $\boldsymbol{y}$ using $\boldsymbol{x} \triangleq (\tilde{\boldsymbol{s}}, \tilde{\boldsymbol{a}}), \tilde{\boldsymbol{a}} \sim \pi_{\boldsymbol{\phi}}(\cdot \mid \tilde{\boldsymbol{s}}); \boldsymbol{y} \triangleq (\boldsymbol{s}, \boldsymbol{a}) \in \mathcal{B}$.
        Calculate the generator loss $\mathcal{L}_g(\boldsymbol{\phi}) = \frac{1}{|\mathcal{B}|} \sum_{\boldsymbol{x}} [\log(1 - D_{\boldsymbol{w}}(\boldsymbol{x}))]$ using $\boldsymbol{x}$ and discriminator $D_{\boldsymbol{w}}$.
        **if** iteration count % $k$ == 0 **then**
            **if** epoch count $< N_{\mathrm{warm}}$ **then**
                Optimize policy with respect to $\mathcal{L}_g(\boldsymbol{\phi})$ only, with learning rate $\eta_{\boldsymbol{\phi}}$, *i.e.*,

$$\arg\min_{\boldsymbol{\phi}} \alpha \cdot \mathcal{L}_g(\boldsymbol{\phi}).$$

            **else**
                Optimize policy with learning rate $\eta_{\boldsymbol{\phi}}$ for the target

$$\arg\min_{\boldsymbol{\phi}} -\frac{1}{|\mathcal{B}|} \sum_{\boldsymbol{s} \sim \mathcal{B}, \boldsymbol{a} \sim \pi_{\boldsymbol{\phi}}(\cdot \mid \boldsymbol{s})} \left[ \min_{j=1,2} Q_{\boldsymbol{\theta}_j}(\boldsymbol{s}, \boldsymbol{a}) \right] + \alpha \cdot \mathcal{L}_g(\boldsymbol{\phi}).$$

            **end if**
        **else**
            Skip the policy improving step.
        **end if**
        {// Training the Discriminator}
        Optimize the discriminator $D_{\boldsymbol{w}}$ to maximize $\frac{1}{|\mathcal{B}|} \sum_{\boldsymbol{y}} [\log D_{\boldsymbol{w}}(\boldsymbol{y})] + \frac{1}{|\mathcal{B}|} \sum_{\boldsymbol{x}} [\log(1 - D_{\boldsymbol{w}}(\boldsymbol{x}))]$ with respect to $\boldsymbol{w}$ with learning rate $\eta_{\boldsymbol{w}}$.
        {// Soft Update the Target Networks}
        $\boldsymbol{\phi}' \leftarrow \beta\boldsymbol{\phi} + (1 - \beta)\boldsymbol{\phi}'; \quad \boldsymbol{\theta}'_j \leftarrow \beta\boldsymbol{\theta}_j + (1 - \beta)\boldsymbol{\theta}'_j$ for $j = 1, 2$.
    **end for**
**end for**

---

# C  PROOFS

We follow the offline RL literature (Liu et al., 2018; Nachum et al., 2019; Kallus & Zhou, 2020; Mousavi et al., 2020; Zhang et al., 2020) to assume the following regularity condition on the MDP

structure, which ensures ergodicity and that the limiting state occupancy measure exists and equals to the stationary distribution of the chain.

**Assumption 1** (Ergodicity of MDP). *The MDP $\mathcal{M}$ is ergodic, i.e., the Markov chains associated with any $\pi_b$ and any $\pi_\phi$ under consideration is positive Harris recurrent (Baxendale, 2011).*

**Lemma 3.** Let $\boldsymbol{A} \in \mathbb{R}^{N \times N}$ be nonsingular and let $\boldsymbol{0} \neq \boldsymbol{b} \in \mathbb{R}^N, \boldsymbol{x} = \boldsymbol{A}^{-1}\boldsymbol{b} \in \mathbb{R}^N$. Let $\Delta \boldsymbol{A} \in \mathbb{R}^{N \times N}$ be an arbitrary perturbation on $\boldsymbol{A}$. Assume that the norm on $\mathbb{R}^{N \times N}$ satisfies $\|\boldsymbol{A}\boldsymbol{x}\| \leq \|\boldsymbol{A}\|\|\boldsymbol{x}\|$ for all $\boldsymbol{A}$ and $\boldsymbol{x}$. If

$$(\boldsymbol{A} + \Delta \boldsymbol{A})(\boldsymbol{x} + \Delta \boldsymbol{x}) = \boldsymbol{b} \text{ and } \frac{\|\Delta \boldsymbol{A}\|}{\|\boldsymbol{A}\|} < \frac{1}{\kappa(\boldsymbol{A})}$$

then

$$\frac{\|\Delta \boldsymbol{x}\|}{\|\boldsymbol{x}\|} \leq \frac{\kappa(\boldsymbol{A})\frac{\|\Delta \boldsymbol{A}\|}{\|\boldsymbol{A}\|}}{1 - \kappa(\boldsymbol{A})\frac{\|\Delta \boldsymbol{A}\|}{\|\boldsymbol{A}\|}} = \frac{\kappa(\boldsymbol{A})}{\frac{\|\boldsymbol{A}\|}{\|\Delta \boldsymbol{A}\|} - \kappa(\boldsymbol{A})}$$

*Proof.* Let $\widehat{\boldsymbol{x}} = \boldsymbol{x} + \Delta \boldsymbol{x}, \boldsymbol{A}(\boldsymbol{x} + \Delta \boldsymbol{x}) + \Delta \boldsymbol{A}\,\widehat{\boldsymbol{x}} = \boldsymbol{b} \implies \boldsymbol{A}\,\Delta \boldsymbol{x} + \Delta \boldsymbol{A}\,\widehat{\boldsymbol{x}} = \boldsymbol{0} \implies \Delta \boldsymbol{x} = -\boldsymbol{A}^{-1}\,\Delta \boldsymbol{A}\,\widehat{\boldsymbol{x}}$. Then,

$$\|\Delta \boldsymbol{x}\| \leq \|\boldsymbol{A}^{-1}\|\|\Delta \boldsymbol{A}\|\,(\|\boldsymbol{x}\| + \|\Delta \boldsymbol{x}\|) = \kappa(\boldsymbol{A})\frac{\|\Delta \boldsymbol{A}\|}{\|\boldsymbol{A}\|}\,(\|\boldsymbol{x}\| + \|\Delta \boldsymbol{x}\|)$$

$$\implies \left(1 - \kappa(\boldsymbol{A})\frac{\|\Delta \boldsymbol{A}\|}{\|\boldsymbol{A}\|}\right)\|\Delta \boldsymbol{x}\| \leq \kappa(\boldsymbol{A})\frac{\|\Delta \boldsymbol{A}\|}{\|\boldsymbol{A}\|}\|\boldsymbol{x}\|$$

$$\implies \frac{\|\Delta \boldsymbol{x}\|}{\|\boldsymbol{x}\|} \leq \frac{\kappa(\boldsymbol{A})\frac{\|\Delta \boldsymbol{A}\|}{\|\boldsymbol{A}\|}}{1 - \kappa(\boldsymbol{A})\frac{\|\Delta \boldsymbol{A}\|}{\|\boldsymbol{A}\|}} = \frac{\kappa(\boldsymbol{A})}{\frac{\|\boldsymbol{A}\|}{\|\Delta \boldsymbol{A}\|} - \kappa(\boldsymbol{A})}$$

since $\kappa(\boldsymbol{A})\,\|\Delta \boldsymbol{A}\| / \|\boldsymbol{A}\| < 1$ by assumption. $\qquad\square$

*Proof of Theorem 1.* By ergodicity, $\boldsymbol{d}_\phi(\boldsymbol{s}), \boldsymbol{d}_b(\boldsymbol{s}) \in \mathbb{R}^N$ uniquely exists. For $\boldsymbol{d} \in \{\boldsymbol{d}_\phi, \boldsymbol{d}_b\}, \boldsymbol{T} \in \{\boldsymbol{T}^\phi, \boldsymbol{T}^b\}$, stationarity implies that $\boldsymbol{d}$ is an eigenvector of $\boldsymbol{T}^\top$ associated with eigenvalue 1, and furthermore

$$\boldsymbol{d}^\top = \boldsymbol{d}^\top \boldsymbol{T} \implies \left(\boldsymbol{I} - \boldsymbol{T}^\top\right)\boldsymbol{d} = \boldsymbol{0} \implies \underbrace{\begin{pmatrix} \boldsymbol{1} \\ \boldsymbol{I} - \boldsymbol{T}^\top \end{pmatrix}}_{\triangleq \widehat{\boldsymbol{T}} \in \mathbb{R}^{(N+1)\times N}} \boldsymbol{d} = \begin{pmatrix} 1 \\ 0 \\ \vdots \\ 0 \end{pmatrix} \tag{10}$$

Since $\boldsymbol{T}$ is a positive matrix, by the Perron-Frobenius theorem (Meyer, 2000), 1 is an eigenvalue of $\boldsymbol{T}^\top$ with algebraic multiplicity, and hence geometry multiplicity, 1. The eigen-equation $\boldsymbol{T}^\top \boldsymbol{v} = 1\boldsymbol{v}$ has unique solution $\boldsymbol{v}$ up to a constant multiplier. Since $\boldsymbol{T}^\top \boldsymbol{d} = \boldsymbol{d}$, the eigenspace of $\boldsymbol{T}^\top$ associated with eigenvalue 1 is $\mathrm{span}(\{\boldsymbol{d}\})$. Hence $\dim \ker\left(\boldsymbol{I} - \boldsymbol{T}^\top\right) = 1 \implies \mathrm{rank}\left(\boldsymbol{I} - \boldsymbol{T}^\top\right) = N - 1 \implies \mathrm{rank}\left(\widehat{\boldsymbol{T}}\right) = N$. The reason is that if $\exists\,\boldsymbol{v} \neq \boldsymbol{0}\ s.t.\ \widehat{\boldsymbol{T}}\,\boldsymbol{v} = \boldsymbol{0}$, then

$$\widehat{\boldsymbol{T}}\boldsymbol{v} = \begin{pmatrix} \boldsymbol{1}\,\boldsymbol{v} \\ (\boldsymbol{I} - \boldsymbol{T}^\top)\,\boldsymbol{v} \end{pmatrix} = \begin{pmatrix} 0 \\ \boldsymbol{0} \end{pmatrix} \implies \boldsymbol{v} = c\,\boldsymbol{d} \text{ for scalar } c \in \mathbb{R} \text{ and } \boldsymbol{1}\,\boldsymbol{v} = c\,\boldsymbol{1}\,\boldsymbol{d} = 0 \implies c = 0,$$

and hence $\boldsymbol{v} = c\,\boldsymbol{d} = \boldsymbol{0}$ which contradicts to $\boldsymbol{v} \neq \boldsymbol{0}$.

Since $\mathrm{rank}\left(\widehat{\boldsymbol{T}}\right) = N, \dim\left(\left\{\boldsymbol{v} \in \mathbb{R}^{N+1} : \boldsymbol{v}^\top \widehat{\boldsymbol{T}} = \boldsymbol{0}\right\}\right) = 1$. For such a $\boldsymbol{v} \neq \boldsymbol{0}, \exists\,i \in \{2, \dots, N+1\}\ s.t.\ v_i \neq 0$. WLOG, assume $v_{N+1} \neq 0$. Let $\boldsymbol{A} \in \mathbb{R}^{N \times N}$ be the first $N$ rows of $\widehat{\boldsymbol{T}}$, then $\mathrm{rank}(\boldsymbol{A}) = N$. The reason is that if $\mathrm{rank}(\boldsymbol{A}) < N, \exists\,\boldsymbol{w} \neq \boldsymbol{0}\ s.t.\ \boldsymbol{w}^\top \boldsymbol{A} = \boldsymbol{0}$, then

$$\begin{pmatrix} \boldsymbol{w} \\ 0 \end{pmatrix}^\top \widehat{\boldsymbol{T}} = \boldsymbol{w}^\top \boldsymbol{A} = \boldsymbol{0}$$

and $v_{N+1} \neq 0 \implies (\boldsymbol{w}^\top, 0)^\top$ is not a constant multiple of $\boldsymbol{v} \implies \dim \ker\left(\widehat{\boldsymbol{T}}^\top\right) \geq 2$, which contradicts to the fact that $\mathrm{rank}\left(\widehat{\boldsymbol{T}}\right) = N$. Thus, we conclude that $\mathrm{rank}(\boldsymbol{A}) = N \implies \boldsymbol{A}$ is invertible.

Let $\boldsymbol{e}^{(1)} = (1, 0, \ldots, 0)^\top \in \mathbb{R}^N$, then Equation 10 implies $\boldsymbol{A}\,\boldsymbol{d} = \boldsymbol{e}^{(1)}$. Plug in $\boldsymbol{d}_\phi, \boldsymbol{d}_b, \boldsymbol{T}_\phi, \boldsymbol{T}_b$ and define $\boldsymbol{A}_\phi, \boldsymbol{A}_b$ similarly, we have $\boldsymbol{A}_\phi\,\boldsymbol{d}_\phi = \boldsymbol{e}^{(1)}, \boldsymbol{A}_b\,\boldsymbol{d}_b = \boldsymbol{e}^{(1)}$. For $\boldsymbol{T}_b$ and $\boldsymbol{T}_\phi$, we notice that by Jensen's inequality,

$$
\begin{aligned}
\sum_{j=1}^{N} \left| (\boldsymbol{T}_b - \boldsymbol{T}_\phi)_{ij} \right| &\leq \sum_{j=1}^{N} \int_{\mathbb{A}} \mathcal{P}\left( S_{t+1} = \boldsymbol{s}^j \mid S_t = \boldsymbol{s}^i, A_t = \boldsymbol{a}_t \right) \left| \pi_b\left( \boldsymbol{a}_t \mid \boldsymbol{s}^i \right) - \pi_\phi\left( \boldsymbol{a}_t \mid \boldsymbol{s}^i \right) \right| d\boldsymbol{a}_t \\
&= \int_{\mathbb{A}} \left( \sum_{j=1}^{N} \mathcal{P}\left( S_{t+1} = \boldsymbol{s}^j \mid S_t = \boldsymbol{s}^i, A_t = \boldsymbol{a}_t \right) \right) \left| \pi_b\left( \boldsymbol{a}_t \mid \boldsymbol{s}^i \right) - \pi_\phi\left( \boldsymbol{a}_t \mid \boldsymbol{s}^i \right) \right| d\boldsymbol{a}_t \\
&= \int_{\mathbb{A}} \left| \pi_b\left( \boldsymbol{a}_t \mid \boldsymbol{s}^i \right) - \pi_\phi\left( \boldsymbol{a}_t \mid \boldsymbol{s}^i \right) \right| d\boldsymbol{a}_t \\
&= \left\| \pi_b\left( \cdot \mid \boldsymbol{s}^i \right) - \pi_\phi\left( \cdot \mid \boldsymbol{s}^i \right) \right\|_1
\end{aligned}
$$

Therefore, by assumption,

$$
\left\| \boldsymbol{T}_b - \boldsymbol{T}_\phi \right\|_\infty = \max_{i=1,\ldots,N} \sum_{j=1}^{N} \left| (\boldsymbol{T}_b - \boldsymbol{T}_\phi)_{ij} \right| \leq \max_{i=1,\ldots,N} \left\| \pi_b\left( \cdot \mid \boldsymbol{s}^i \right) - \pi_\phi\left( \cdot \mid \boldsymbol{s}^i \right) \right\|_1 \leq \epsilon.
$$

Let $\boldsymbol{A}_\phi = \boldsymbol{A}_b + \Delta\boldsymbol{A}$. For $\|\Delta\boldsymbol{A}\|_1$, we notice that

$$
\begin{aligned}
\|\Delta\boldsymbol{A}\|_1 = \|\boldsymbol{A}_\phi - \boldsymbol{A}_b\|_1 &\leq \left\| \left( \boldsymbol{I} - \boldsymbol{T}_\phi^\top \right) - \left( \boldsymbol{I} - \boldsymbol{T}_b^\top \right) \right\|_1 \\
&= \left\| \left( \boldsymbol{T}_b^\top - \boldsymbol{T}_\phi^\top \right) \right\|_1 = \left\| (\boldsymbol{T}_b - \boldsymbol{T}_\phi)^\top \right\|_1 = \|\boldsymbol{T}_b - \boldsymbol{T}_\phi\|_\infty \leq \epsilon.
\end{aligned}
$$

Notice that matrix 1-norm satisfies $\|\boldsymbol{M}\boldsymbol{v}\|_1 \leq \|\boldsymbol{M}\|_1 \|\boldsymbol{v}\|_1$ for all matrix $\boldsymbol{M}$ and vector $\boldsymbol{v}$, that $\|\boldsymbol{A}_b\|_1 \geq 1$ and that $\|\boldsymbol{d}_b\|_1 = 1$. Lemma 3 implies that

$$
\begin{aligned}
\frac{\|\Delta\boldsymbol{d}\|_1}{\|\boldsymbol{d}_b\|_1} = \|\Delta\boldsymbol{d}\|_1 = \|\boldsymbol{d}_\phi - \boldsymbol{d}_b\|_1 \\
&\leq \frac{\kappa(\boldsymbol{A}_b)}{\frac{1}{\|\Delta\boldsymbol{A}\|_1} - \kappa(\boldsymbol{A}_b)} \\
&\leq \frac{\kappa(\boldsymbol{A}_b)}{\frac{1}{\epsilon} - \kappa(\boldsymbol{A}_b)} = \frac{\epsilon\,\kappa(\boldsymbol{A}_b)}{1 - \epsilon\,\kappa(\boldsymbol{A}_b)} \\
&\leq \frac{\epsilon\,\kappa_{max}}{1 - \epsilon\,\kappa_{max}} \to 0 \quad \text{as } \epsilon \to 0.
\end{aligned}
\tag{11}
$$

$\square$

*Proof of Theorem 2.* By ergodicity, $\boldsymbol{d}_\phi(\boldsymbol{s}), \boldsymbol{d}_{b_k}(\boldsymbol{s}) \in \mathbb{R}^N$ uniquely exists, $\forall k$. For $\boldsymbol{d} \in \{\boldsymbol{d}_\phi, \boldsymbol{d}_{b_1}, \ldots, \boldsymbol{d}_{b_k}\}$ and $\boldsymbol{T} \in \{\boldsymbol{T}^\phi, \boldsymbol{T}^{b_1}, \ldots, \boldsymbol{T}^{b_K}\}$, we follow the steps and notations in the proof of Theorem 1 to conclude that $\mathrm{rank}\,(\boldsymbol{A}) = N \implies \boldsymbol{A}$ is invertible and that $\boldsymbol{A}\boldsymbol{d} = \boldsymbol{e}^{(1)} \in \mathbb{R}^N$. Plug in $\boldsymbol{d}_\phi, \boldsymbol{d}_{b_k}, \boldsymbol{T}_\phi, \boldsymbol{T}_{b_k}$ and define $\boldsymbol{A}_\phi, \boldsymbol{A}_{b_k}$ similarly, we have $\boldsymbol{A}_\phi\boldsymbol{d}_\phi = \boldsymbol{e}^{(1)}, \boldsymbol{A}_{b_k}\boldsymbol{d}_{b_k} = \boldsymbol{e}^{(1)}$. For the transition matrix $\boldsymbol{T}_b$ induced by the mixture of policies $\pi_b$, we have

$$
\begin{aligned}
\boldsymbol{T}_{b,(i,j)} = p_{\pi_b}\left( S_{t+1} = \boldsymbol{s}^j \mid S_t = \boldsymbol{s}^i \right) &= \int_{\mathbb{A}} \mathcal{P}\left( S_{t+1} = \boldsymbol{s}^j \mid S_t = \boldsymbol{s}^i, A_t = \boldsymbol{a}_t \right) \pi_b\left( \boldsymbol{a}_t \mid \boldsymbol{s}^i \right) d\boldsymbol{a}_t \\
&= \sum_{k=1}^{K} w_k \int_{\mathbb{A}} \mathcal{P}\left( S_{t+1} = \boldsymbol{s}^j \mid S_t = \boldsymbol{s}^i, A_t = \boldsymbol{a}_t \right) \pi_{b_k}\left( \boldsymbol{a}_t \mid \boldsymbol{s}^i \right) d\boldsymbol{a}_t = \sum_{k=1}^{K} w_k \boldsymbol{T}_{b_k,(i,j)}
\end{aligned}
$$

and therefore $\boldsymbol{T}_b = \sum_{k=1}^{K} w_k \boldsymbol{T}_{b_k}$.

For $\boldsymbol{T}_{b_k}$ and $\boldsymbol{T}_\phi$, as in the proof of Theorem 1, by Jensen's inequality,

$$
\sum_{j=1}^{N} \left| (\boldsymbol{T}_{b_k} - \boldsymbol{T}_\phi)_{ij} \right| \leq \left\| \pi_{b_k}\left( \cdot \mid \boldsymbol{s}^i \right) - \pi_\phi\left( \cdot \mid \boldsymbol{s}^i \right) \right\|_1
$$

Therefore, by assumption,

$$\|\boldsymbol{T}_{b_k} - \boldsymbol{T}_\phi\|_\infty = \max_{i=1,\dots,N} \sum_{j=1}^N \left|(\boldsymbol{T}_{b_k} - \boldsymbol{T}_\phi)_{ij}\right| \le \max_{i=1,\dots,N} \left\|\pi_{b_k}\left(\cdot \mid \boldsymbol{s}^i\right) - \pi_\phi\left(\cdot \mid \boldsymbol{s}^i\right)\right\|_1 \le \epsilon.$$

Let $\boldsymbol{A}_\phi = \boldsymbol{A}_{b_k} + \Delta\boldsymbol{A}_{b_k}$. For $\|\Delta\boldsymbol{A}_{b_k}\|_1$, we have

$$\|\Delta\boldsymbol{A}_{b_k}\|_1 = \|\boldsymbol{A}_\phi - \boldsymbol{A}_{b_k}\|_1 \le \left\|\boldsymbol{T}_{b_k}^\top - \boldsymbol{T}_\phi^\top\right\|_1 = \|\boldsymbol{T}_{b_k} - \boldsymbol{T}_\phi\|_\infty \le \epsilon.$$

Note that $\forall k, \|\boldsymbol{A}_{b_k}\|_1 \ge 1 + 1 - \sum_{i=1}^N \boldsymbol{T}_{1i}^{b_k} = 1, \|\boldsymbol{d}_{b_k}\|_1 = 1$. Lemma 3 implies that

$$\frac{\|\Delta\boldsymbol{d}_k\|_1}{\|\boldsymbol{d}_{b_k}\|_1} = \|\Delta\boldsymbol{d}_k\|_1 = \|\boldsymbol{d}_\phi - \boldsymbol{d}_{b_k}\|_1 \le \frac{\epsilon\,\kappa\left(\boldsymbol{A}_{b_k}\right)}{1 - \epsilon\,\kappa\left(\boldsymbol{A}_{b_k}\right)} \le \frac{\epsilon\,\kappa_{max.k}}{1 - \epsilon\,\kappa_{max,k}}$$

Thus for the relative distance between $\boldsymbol{d}_\phi$ and $\boldsymbol{d}_b$, we have

$$\|\boldsymbol{d}_b\|_1 = \sum_{i=1}^N \boldsymbol{d}_b\left(\boldsymbol{s}^i\right) = \sum_{i=1}^N \sum_{k=1}^K w_k \boldsymbol{d}_{b_k}\left(\boldsymbol{s}^i\right) = \sum_{k=1}^K w_k \sum_{i=1}^N \boldsymbol{d}_{b_k}\left(\boldsymbol{s}^i\right) = \sum_{k=1}^K w_k = 1,$$

$$\frac{\|\boldsymbol{d}_\phi - \boldsymbol{d}_b\|_1}{\|\boldsymbol{d}_b\|_1} = \|\boldsymbol{d}_\phi - \boldsymbol{d}_b\|_1 = \left\|\sum_{k=1}^K \left(w_k \boldsymbol{d}_\phi - w_k \boldsymbol{d}_{b_k}\right)\right\|_1$$

$$\le \sum_{k=1}^K w_k \|\boldsymbol{d}_\phi - \boldsymbol{d}_{b_k}\|_1 \le \sum_{k=1}^K w_k \frac{\epsilon\,\kappa_{max.k}}{1 - \epsilon\,\kappa_{max,k}} \le \frac{\epsilon\,\kappa_{max}}{1 - \epsilon\,\kappa_{max}} \to 0, \quad \text{as } \epsilon \to 0.$$

Plug $w_k = 1/K, \forall k \in \{1, \dots, K\}$ into the second to last equation, we get

$$\|\boldsymbol{d}_\phi - \boldsymbol{d}_b\|_1 \le \frac{1}{K}\sum_{k=1}^K \frac{\epsilon\,\kappa_{max,k}}{1 - \epsilon\,\kappa_{max,k}} \to 0, \quad \text{as } \epsilon \to 0.$$

$\square$

*Remark.* In general, $\boldsymbol{d}_b^\top \boldsymbol{T}_b = \sum_{k=1}^K w_k^2 \boldsymbol{d}_{b_k}^\top + \sum_{i\neq j} w_i w_j \boldsymbol{d}_{b_i}^\top \boldsymbol{T}^{b_j} \neq \boldsymbol{d}_b^\top$. One sufficient condition is $\boldsymbol{d}_{b_i}^\top \boldsymbol{T}^{b_j} = \boldsymbol{d}_{b_i}^\top, \forall i,j \implies \boldsymbol{d}_{b_i} = \boldsymbol{d}_{b_j}, \forall i,j \implies \pi_{b_i} = \pi_{b_j}, \forall i,j$, similar to Ho & Ermon (2016). In such case, $\pi_b$ reduces to a single policy, not a mixture, and Theorem 1 applies.

# D    TECHNICAL DETAILS

## D.1    TOY EXPERIMENT

Denote the total sample size as $N_{\text{total}}$, we follow the convention to construct the eight-Gaussian dataset as in Algorithm 3. In our experiment we use $N_{\text{total}} = 2000$.

---

**Algorithm 3** Constructing the Eight-Gaussian Dataset

---

**Input:** Total sample size $N_{\text{total}}$.
**Output:** Generated dataset $\mathbb{D}_{\text{Gaussian}}$.
**while** Dataset size $< N_{\text{total}}$ **do**
    Draw a random center $\boldsymbol{c}$ uniformly

$$\boldsymbol{c} \sim \left\{\left(\sqrt{2}, 0\right), \left(-\sqrt{2}, 0\right), \left(0, \sqrt{2}\right), \left(0, -\sqrt{2}\right), (1, 1), (1, -1), (-1, 1), (-1, -1)\right\}.$$

    Sample datapoint $\boldsymbol{x} = (x, y) \sim \mathcal{N}\left(\boldsymbol{c}, 2 \times 10^{-4} \cdot \boldsymbol{I}_2\right)$.
    Store $\boldsymbol{x}$ in the dataset.
**end while**

---

We are interested in the 2-D eight-Gaussian dataset because **(a)** the conditional distribution of $p(y \mid x)$ is multi-modal in many $x$; and **(b)** interpolation is needed to fill-in the blanks between Gaussian-centers, where a smooth-interpolation into a circle is naturally expected.

To rephrase this dataset into offline reinforcement learning setting, we define $x$ as state and the corresponding $y$ as action. Note that in the behavior cloning algorithm, the information of reward, next state, and the episodic termination is not required. Hence, the generated dataset $\mathbb{D}_{\text{Gaussian}}$ can serve as an offline RL dataset readily applicable to train behavior cloning policies.

In order to compare the ability to approximate the behavior policy by the KL loss and the JSD loss, the Gaussian policy and the implicit policy, the conditional-distribution matching and the joint-visitation matching, we fit a conditional VAE ("CVAE"), a Gaussian generator conditional GAN ("G-CGAN") and a conditional GAN ("CGAN") using the conditional-distribution matching approach similar to Kumar et al. (2019). We fit, using basic joint-visitation matching strategy, a conditional GAN ("GAN"). As discussed in Appendix A, the major distinction between "CGAN" and "GAN" is that the former uses the same states in constructing the generator samples and the data samples while the later resamples states.

The network architecture of our conditional VAE is as follows.

Conditional Variational Auto-encoder (CVAE) in Toy Experiment

Encoder
```
Linear(state_dim+action_dim, H)
BatchNorm1d(H)
ReLU
Linear(H, H//2)
BatchNorm1d(H//2)
ReLU
mean = Linear(H//2, latent_dim)
log_std = Linear(H//2, latent_dim)
```

Decoder
```
Linear(state_dim+latent_dim, H)
BatchNorm1d(H)
ReLU
Linear(H, H//2)
BatchNorm1d(H//2)
ReLU
Linear(H//2, action_dim)
```

with hidden dimension `H` $= 100$ and latent dimension `latent_dim` $= 50$. CVAE is trained for 1200 epochs with a mini-batch size of 100 and random seed 0, using the mean-squared-error as the reconstruction loss, and the Gaussian-case closed-form formula in Kingma & Welling (2013) for the KL term.

The network architecture of our conditional GAN, used in "CGAN" and "GAN," is as follows.

Conditional Generative Adversarial Nets (CGAN) in Toy Experiment

Generator
```
Linear(state_dim+z_dim, H)
BatchNorm1d(H)
ReLU
Linear(H, H//2)
BatchNorm1d(H//2)
ReLU
Linear(H//2, action_dim)
```

Discriminator
```
Linear(state_dim+action_dim, H)
LeakyReLU(0.1)
Linear(H, H//2)
LeakyReLU(0.1)
Linear(H//2, 1)
```

where the structure of `BatchNorm1d, LeakyReLU` follows Radford et al. (2016). Here we again use `H` $= 100,$ `z_dim` $= 50$. Conditional GAN is trained for 2000 epochs with a mini-batch size of 100 and random seed 0. We follow Radford et al. (2016) to train CGAN using Adam optimizer with $\beta_1 = 0.5$.

The network architecture of our Gaussian-generator version of conditional GAN, used in the experiments of Appendix F, is as follows.

```
Generator
Linear(state_dim, H)
BatchNorm1d(H)
ReLU
Linear(H, H//2)
BatchNorm1d(H//2)
ReLU
mean = Linear(H//2, action_dim), log_std = Linear(H//2, action_dim)
```

with Discriminator and other technical details the same as CGAN. This Gaussian-generator version of CGAN is again trained for 2000 epochs with a mini-batch size of 100, random seed 0, and $\beta_1 = 0.5$ in the Adam optimizer.

Our test set is formed by a random sample of 2000 new states ($x$) from $[-1.5, 1.5]$ together with the states in the training set. The performance on the test set thus shows both the concentration on the eight centers and the smooth interpolation between centers, which translates into a good and smooth fit to the behavior policy. Figure 1 shows the training set ("Truth") and the kernel-density-estimate plot of each methods.

### D.2  REINFORCEMENT LEARNING EXPERIMENTS

In practice, the rollouts contained in the offline dataset have finite horizon, and thus special treatment is needed per appearance of the terminal states in calculating the Bellman update target. We follow the standard treatment (Mnih et al., 2013; Sutton & Barto, 2018) to define the update target $y$ as

$$\widetilde{Q}\left(\boldsymbol{s}, \boldsymbol{a}\right) = \begin{cases} r\left(\boldsymbol{s}, \boldsymbol{a}\right) + \gamma \widetilde{Q}'\left(\boldsymbol{s}', \boldsymbol{a}'\right) & \text{if } \boldsymbol{s}' \text{ is a non-terminal state} \\ r\left(\boldsymbol{s}, \boldsymbol{a}\right) & \text{if } \boldsymbol{s}' \text{ is a terminal state} \end{cases},$$

where $\widetilde{Q}'\left(\boldsymbol{s}', \boldsymbol{a}'\right)$ refers to the expectation term in Equation 4 for basic algorithm (Section 4.1) or the expectation term in Equation 8 for the enhanced versions with state-smoothing at the Bellman Backup (Section 4.2).

For simplicity, we follow White (2016) to choose the noise distribution $p_{\boldsymbol{z}}\left(\boldsymbol{z}\right)$ as the multivariate standard normal distribution, where the dimension of $\boldsymbol{z}$ is conveniently chosen as $\dim\left(\boldsymbol{z}\right) = \min(10, \texttt{state\_dim}//2)$. To sample from the implicit policy, for each state $\boldsymbol{s}$, we first sample independently $\boldsymbol{z} \sim \mathcal{N}\left(\boldsymbol{0}, \boldsymbol{I}\right)$. We then concatenate $\boldsymbol{s}$ with $\boldsymbol{z}$ and feed the resulting $[\boldsymbol{s}, \boldsymbol{z}]$ into the deterministic policy network to generate stochastic actions. To sample from a small region around the next state $\boldsymbol{s}'$ (Section 4.2), we keep the original $\boldsymbol{s}'$ and repeat it additionally $N_B$ times. For each of the $N_B$ replications, we add an independent Gaussian noise $\boldsymbol{\epsilon} \sim \mathcal{N}(\boldsymbol{0}, \sigma_B^2 \boldsymbol{I})$. The original $\boldsymbol{s}'$ and its $N_B$ noisy replications are then fed into the implicit policy to sample the corresponding action.

For fair comparison, we use the same network architecture as the official implementation of the BCQ algorithm, which will be stated in detail in Sections D.2.1. Due to limited computational resources, we leave a fine-tuning of the noise distribution $p_{\boldsymbol{z}}\left(\boldsymbol{z}\right)$, the network architectures, and the optimization hyperparameters for future work, which also leaves room for further improving our results.

For a more stable training of the policy, we adopt the warm start strategy (Kumar et al., 2020; Yue et al., 2020). Specifically, in the first $N_{\text{warm}}$ epochs, the policy is trained to minimize $\mathcal{L}_g$ only, with the same learning rate $\eta_{\boldsymbol{\phi}}$ as the rest epochs that also try to maximize the expected Q-values.

**Datasets.** We use the continuous control tasks provided by the D4RL dataset (Fu et al., 2021) to conduct algorithmic evaluations. Due to limited computational resources, we select therein the "medium-expert," "medium-replay," and "medium" datasets for the Hopper, HalfCheetah, Walker2d tasks in the Gym-MuJoCo domain, which are commonly used benchmarks in prior work (Fujimoto et al., 2019; Kumar et al., 2019; Wu et al., 2019; Kumar et al., 2020). We follow the literature (Cang et al., 2021; Chen et al., 2021; Kostrikov et al., 2021) to not test on the "random" and "expert" datasets as they are less practical (Matsushima et al., 2021) and can be respectively solved by directly using standard off-policy RL algorithms (Agarwal et al., 2020) and the behavior cloning algorithms. We note that a comprehensive benchmarking of prior offline-RL algorithms on the "expert" datasets is currently unavailable in the literature, which is out of the scope of this paper. Apart from the Gym-MuJoCo domain, we also consider the Maze2D domain[2] of tasks for the non-Markovian data-collecting policy and the Adroit tasks[3] (Rajeswaran et al., 2018) for their sparse reward-signal and high dimensionality.

**Evaluation Protocol.** In all the experiments, we follow Fu et al. (2021) to use the "v0" version of the datasets in the Gym-MuJoCo and Adroit domains. In our preliminary study, we find that the rollout results of some existing algorithms can be unstable across epochs in some datasets, even

---

[2]We use the tasks "maze2d-umaze," "maze2d-medium," and "maze2d-large."
[3]We use the tasks "pen-human," "pen-cloned," "pen-expert," and "door-expert."

towards the end of training. To reduce the randomness in evaluation, we report, for our algorithm, the mean and standard deviation of the last five rollouts across three random seeds $\{0, 1, 2\}$. We train all agents for 1000 epochs, where each epoch consists of 1000 mini-batch stochastic gradient descent steps. We rollout each agent for 10 episodes after each epoch of training.

### D.2.1 GAN JOINT MATCHING

In approximately matching the Jensen-Shannon divergence between the state-action visitation frequencies via Generative Adversarial Nets, a crucial step is to stably and effectively train the GAN structure. To this end, we adopt the following tricks from the literature.

- To provide stronger gradients early in training, rather than training the policy $\pi_\phi$ to minimize

$$\mathbb{E}_{\boldsymbol{x}} \left[ \log \left( 1 - D_{\boldsymbol{w}}(\boldsymbol{x}) \right) \right]$$

  we follow (Goodfellow et al., 2014) to train $\pi_\phi$ to maximize

$$\mathbb{E}_{\boldsymbol{x}} \left[ \log \left( D_{\boldsymbol{w}}(\boldsymbol{x}) \right) \right]$$

- Motivated by (Radford et al., 2016), we use LeakyReLU activation in both the generator and discriminator, with default `negative_slope=0.01`.
- To stabilize the training, we follow (Radford et al., 2016) to use a reduced momentum term $\beta_1 = 0.4$ in the Adam optimizer (Kingma & Ba, 2014).
- We follow (Radford et al., 2016) to use actor and discriminator learning rate $\eta_\phi = \eta_{\boldsymbol{w}} = 2 \times 10^{-4}$.
- To avoid overfitting of the discriminator, we are motivated by Salimans et al. (2016) and Goodfellow (2017) to use one-sided label smoothing with soft and noisy labels. Specifically, the labels for the data sample $\boldsymbol{y}$ is replaced with a random number between $0.8$ and $1.0$, instead of the original $1$. No label smoothing is applied for the generator sample $\boldsymbol{x}$, therefore their labels are all $0$.

  The loss function for training the discriminator in GAN is the Binary Cross Entropy between the labels and the outputs from the discriminator.

Furthermore, motivated by TD3 (Fujimoto et al., 2018) and GAN (Section 2), we update $\pi_\phi(\cdot \mid \boldsymbol{s})$ once per $k$ updates of the critics and discriminator.

Table 2 shows the hyperparameters for our GAN joint-matching framework. Note that several simplifications are made to minimize hyperparameter tuning, such as fixing $\eta_\phi = \eta_{\boldsymbol{w}}$ as in (Radford et al., 2016) and $\sigma_B = \sigma_J$. We also fix $N_a = 1$ to ease computation.

We comment that many of these hyperparameters can be set based on literature, for example, we use $\eta_\phi = \eta_{\boldsymbol{w}} = 2 \times 10^{-4}$ as in Radford et al. (2016), $\eta_{\boldsymbol{\theta}} = 3 \times 10^{-4}$ and $N_{\text{warm}} = 40$ as in Kumar et al. (2020), $\lambda = 0.75$ as in Fujimoto et al. (2019) and policy frequency $k = 2$ as in Fujimoto et al. (2018). Unless specified, the same hyperparameter is used across all datasets.

Below we state the network architectures of the actor, critic, and the discriminator in GAN. Note that we use a pair of critic networks with the same architecture to perform clipped double Q-learning.

Actor
```
Linear(state_dim+noise_dim, 400)
LeakyReLU
Linear(400, 300)
LeakyReLU
Linear(300, action_dim)
max_action * tanh
```

Critic
```
Linear(state_dim+action_dim, 400)
LeakyReLU
Linear(400, 300)
LeakyReLU
Linear(300, 1)
```

Discriminator in GAN
```
Linear(state_dim+action_dim, 400)
LeakyReLU
Linear(400, 300)
LeakyReLU
Linear(300, 1)
Sigmoid
```

Note that the network architectures follow from Fujimoto et al. (2019) and that all the LeakyReLU activation uses the default `negative_slope=0.01`.

Table 2: Default Hyperparameters for GAN joint matching.

| Hyperparameter | Value |
|---|---|
| Optimizer | Adam Kingma & Ba (2014) |
| Learning rate $\eta_{\theta}$ | $3 \times 10^{-4}$ |
| Learning rate $\eta_{\phi}, \eta_{w}$ | $2 \times 10^{-4}$ |
| Log Lagrange multiplier $\log \alpha$ for non-Adroit datasets | 4.0 |
| Log Lagrange multiplier $\log \alpha$ for Adroit datasets | 8.0 |
| Evaluation frequency | $10^3$ |
| Training iterations | $10^6$ |
| Batch size | 512 |
| Discount factor | 0.99 |
| Target network update rate $\beta$ | 0.005 |
| Weighting for clipped double Q-learning $\lambda$ | 0.75 |
| Noise distribution $p_{\boldsymbol{z}}(\boldsymbol{z})$ | $\mathcal{N}(\boldsymbol{0}, \boldsymbol{I})$ |
| Standard deviations for state smoothing $\sigma_B, \sigma_J$ | $3 \times 10^{-4}$ |
| Number of smoothed states in Bellman backup $N_B$ | 50 |
| Number of actions $\hat{\boldsymbol{a}}$ at each $\hat{\boldsymbol{s}}$ $N_a$ | 1 |
| Number of epochs for warm start $N_{\text{warm}}$ | 40 |
| Policy frequency $k$ | 2 |
| Random seeds | $\{0, 1, 2\}$ |

### D.2.2 RESULTS OF CQL

We note that the official CQL GitHub repository does not provide hyperparameter settings for the Maze2D and Adroit domain of tasks. For datasets in these two domains, we train an CQL agent using five hyperparameter settings: four recommended Gym-MuJoCo settings and one recommended Ant-Maze setting. We then calculate the average (normalized) return of the last five rollouts over random seeds $\{0, 1, 2\}$ and report the per-dataset best results across those five hyperparameter settings. We comment that this approach is non-conventional, but rather a compensation for the missing of recommended hyperparameter settings, and may give CQL some advantage on the Maze2D and Adroit domains. For the Gym-MuJoCo domain, we follow Kumar et al. (2020) to use the recommended hyperparameter setting across all datasets.

### D.2.3 ABLATION STUDY ON GAUSSIAN POLICY

The network architecture of the Gaussian policy we used in the ablation study (Section 5.2) follows common practice ((Haarnoja et al., 2018a; Kumar et al., 2020)).

```
Gaussian Policy
Linear(state_dim, 400)
LeakyReLU
Linear(400, 300)
LeakyReLU
mean = Linear(300, action_dim)
log_std = Linear(300, action_dim)
```

Critics and discriminator are the same as in the implicit policy case (Appendix D.2.1).

For action selection from the Gaussian policy, a given state $\boldsymbol{s}$ is first mapped to the mean $\boldsymbol{\mu}(\boldsymbol{s})$ and standard deviation vector $\boldsymbol{\sigma}(\boldsymbol{s})$. A raw action is sampled as $\boldsymbol{a}_{\text{raw}} \sim \mathcal{N}\left(\boldsymbol{\mu}(\boldsymbol{s}), \text{diag}\left(\boldsymbol{\sigma}^2(\boldsymbol{s})\right)\right)$. Finally, $\boldsymbol{a}_{\text{raw}}$ is mapped into the action space as $\text{max\_action} \times \tanh(\boldsymbol{a}_{\text{raw}})$.

For fair comparison, other technical details, including the training procedure and hyperparameter setting, are exactly the same as the implicit policy case (Appendix D.2.1).

# E ADDITIONAL TABLES

## E.1 RAW SCORES OF THE MAIN TABLE

In Table 3 we report the un-normalized results corresponding to Table 1 for reference.

Table 3: Raw returns for experiments on the D4RL suite of tasks. We performs experiments on tasks from the Gym-Mojoco, Maze2D, and Adroit domains. For our algorithm, we report in this table the mean and standard deviation of the raw returns of the last five rollouts across three random seeds $\{0, 1, 2\}$. We run CQL ourselves and report the average raw return of the last five rollouts over random seeds $\{0, 1, 2\}$. Results for other algorithms are from Fu et al. (2021).

| Task Name | SAC-off | BEAR | BRAC-v | BCQ | CQL | GAN-Cond:Basic | GAN-Joint:Basic | GAN-Joint |
|---|---|---|---|---|---|---|---|---|
| maze2d-umaze | 145.6 | 28.6 | 1.7 | 41.5 | 93.5 | $96.0 \pm 27.1$ | $103.3 \pm 15.2$ | $79.2 \pm 23.3$ |
| maze2d-medium | 82.0 | 89.8 | 102.4 | 35.0 | 94.3 | $125.8 \pm 48.0$ | $117.3 \pm 27.3$ | $197.1 \pm 67.7$ |
| maze2d-large | 1.5 | 19.0 | 115.2 | 23.2 | 123.5 | $105.4 \pm 47.7$ | $145.8 \pm 55.5$ | $197.3 \pm 69.4$ |
| halfcheetah-medium | -808.6 | 4897.0 | 5473.8 | 4767.9 | 4566.2 | $5155.3 \pm 30.8$ | $5149.0 \pm 61.6$ | $5197.4 \pm 31.6$ |
| walker2d-medium | 44.2 | 2717.0 | 3725.8 | 2441.0 | 2763.2 | $3009.3 \pm 312.3$ | $3212.9 \pm 416.3$ | $3181.2 \pm 396.9$ |
| hopper-medium | 5.7 | 1674.5 | 990.4 | 1752.4 | 1103.6 | $2175.0 \pm 692.5$ | $2142.5 \pm 499.6$ | $1935.2 \pm 887.4$ |
| halfcheetah-medium-replay | -581.3 | 4517.9 | 5640.6 | 4463.9 | 5105.5 | $3726.5 \pm 295.0$ | $3599.9 \pm 342.3$ | $3832.3 \pm 281.1$ |
| walker2d-medium-replay | 87.8 | 883.8 | 44.5 | 688.7 | 755.0 | $318.6 \pm 107.4$ | $458.0 \pm 91.1$ | $471.6 \pm 110.0$ |
| hopper-medium-replay | 93.3 | 1076.8 | -0.8 | 1057.8 | 941.3 | $812.1 \pm 51.8$ | $1166.9 \pm 168.8$ | $938.9 \pm 82.3$ |
| halfcheetah-medium-expert | -55.7 | 6349.6 | 4926.6 | 7750.8 | 4000.7 | $9150.7 \pm 1142.8$ | $8936.9 \pm 760.8$ | $9132.9 \pm 1259.3$ |
| walker2d-medium-expert | -5.1 | 1842.7 | 3747.5 | 2640.3 | 3666.6 | $3367.5 \pm 631.3$ | $3513.0 \pm 691.8$ | $3269.7 \pm 1008.6$ |
| hopper-medium-expert | 32.9 | 3113.5 | 5.1 | 3588.5 | 3347.8 | $2193.1 \pm 693.5$ | $2286.4 \pm 870.9$ | $3232.2 \pm 944.3$ |
| pen-human | 284.8 | 66.3 | 114.7 | 2149.0 | 159.8 | $1672.2 \pm 473.3$ | $1915.7 \pm 407.0$ | $1451.9 \pm 729.7$ |
| pen-cloned | 797.6 | 885.4 | 22.2 | 1407.8 | 139.8 | $675.1 \pm 431.2$ | $1028.4 \pm 639.7$ | $633.3 \pm 430.5$ |
| pen-expert | 277.4 | 3254.1 | 6.4 | 3521.3 | 2954.5 | $3855.7 \pm 528.6$ | $3943.6 \pm 429.0$ | $4302.1 \pm 441.4$ |
| door-expert | 163.8 | 2980.1 | -66.6 | 2850.7 | 2525.8 | $2904.6 \pm 110.3$ | $3001.0 \pm 106.1$ | $2979.9 \pm 108.9$ |

## E.2 TABLES FOR THE ABLATION STUDY

Table 4 - 7 correspond to the results for our ablation study in Section 5.2.

Table 4: Normalized returns for comparing the implicit policy with the Gaussian policy on the basic algorithm (Section 4.1) on the D4RL suite of tasks. The reported number are the means and standard deviations of the normalized returns of the last five rollouts across three random seeds $\{0, 1, 2\}$.

| Task Name | GAN-Joint-Matching: Basic | GAN-Joint-Matching: Basic, Gaussian Policy |
|---|---|---|
| maze2d-umaze | $57.6 \pm 11.0$ | $23.9 \pm 8.1$ |
| maze2d-medium | $39.4 \pm 10.3$ | $0.9 \pm 7.7$ |
| maze2d-large | $52.1 \pm 20.8$ | $3.3 \pm 0.7$ |
| halfcheetah-medium | $43.7 \pm 0.5$ | $43.8 \pm 0.4$ |
| walker2d-medium | $70.0 \pm 9.1$ | $51.6 \pm 10.9$ |
| hopper-medium | $66.5 \pm 15.4$ | $79.8 \pm 14.1$ |
| halfcheetah-medium-replay | $31.3 \pm 2.8$ | $33.0 \pm 3.1$ |
| walker2d-medium-replay | $9.9 \pm 2.0$ | $9.1 \pm 1.7$ |
| hopper-medium-replay | $36.5 \pm 5.2$ | $26.3 \pm 2.2$ |
| halfcheetah-medium-expert | $74.2 \pm 6.1$ | $76.8 \pm 6.7$ |
| walker2d-medium-expert | $76.5 \pm 15.1$ | $59.6 \pm 21.9$ |
| hopper-medium-expert | $70.9 \pm 26.8$ | $84.6 \pm 10.6$ |
| pen-human | $61.0 \pm 13.7$ | $55.7 \pm 21.1$ |
| pen-cloned | $31.3 \pm 21.5$ | $36.6 \pm 14.6$ |
| pen-expert | $129.1 \pm 14.4$ | $118.8 \pm 15.1$ |
| door-expert | $104.1 \pm 3.6$ | $39.5 \pm 24.6$ |
| Average Score | **59.6** | 46.5 |

Table 5: Normalized returns for comparing our full algorithm with its counterpart of no state-smoothing in the joint matching of the state-action-visitation. The reported number are the means and standard deviations of the normalized returns of the last five rollouts across three random seeds $\{0, 1, 2\}$.

| Task Name | GAN-Joint-Matching: Full Algorithm | GAN-Joint-Matching: No State-smoothing in Joint Matching |
|---|---|---|
| maze2d-umaze | $40.1 \pm 16.9$ | $47.6 \pm 4.7$ |
| maze2d-medium | $69.6 \pm 25.6$ | $35.4 \pm 5.5$ |
| maze2d-large | $71.3 \pm 26.0$ | $69.9 \pm 26.5$ |
| halfcheetah-medium | $44.1 \pm 0.3$ | $44.0 \pm 0.4$ |
| walker2d-medium | $69.3 \pm 8.6$ | $62.7 \pm 8.6$ |
| hopper-medium | $60.1 \pm 27.3$ | $81.8 \pm 16.6$ |
| halfcheetah-medium-replay | $33.1 \pm 2.3$ | $31.8 \pm 2.7$ |
| walker2d-medium-replay | $10.2 \pm 2.4$ | $9.9 \pm 2.5$ |
| hopper-medium-replay | $29.5 \pm 2.5$ | $29.5 \pm 2.4$ |
| halfcheetah-medium-expert | $75.8 \pm 10.1$ | $69.1 \pm 8.3$ |
| walker2d-medium-expert | $71.2 \pm 22.0$ | $73.8 \pm 18.4$ |
| hopper-medium-expert | $99.9 \pm 29.0$ | $67.8 \pm 18.9$ |
| pen-human | $45.5 \pm 24.5$ | $54.3 \pm 19.6$ |
| pen-cloned | $18.0 \pm 14.4$ | $22.5 \pm 17.4$ |
| pen-expert | $141.1 \pm 14.8$ | $142.4 \pm 14.1$ |
| door-expert | $103.4 \pm 3.7$ | $102.5 \pm 3.2$ |
| Average Score | **61.4** | 59.1 |

Table 6: Normalized returns for comparing our full algorithm with its counterpart of no state-smoothing in the Bellman backup. The reported number are the means and standard deviations of the normalized returns of the last five rollouts across three random seeds $\{0, 1, 2\}$.

| Task Name | GAN-Joint-Matching: Full Algorithm | GAN-Joint-Matching: No State-smoothing in Bellman Backup |
|---|---|---|
| maze2d-umaze | $40.1 \pm 16.9$ | $60.7 \pm 18.3$ |
| maze2d-medium | $69.6 \pm 25.6$ | $51.2 \pm 10.3$ |
| maze2d-large | $71.3 \pm 26.0$ | $50.8 \pm 6.7$ |
| halfcheetah-medium | $44.1 \pm 0.3$ | $44.1 \pm 0.4$ |
| walker2d-medium | $69.3 \pm 8.6$ | $60.7 \pm 11.0$ |
| hopper-medium | $60.1 \pm 27.3$ | $66.9 \pm 22.9$ |
| halfcheetah-medium-replay | $33.1 \pm 2.3$ | $33.4 \pm 2.7$ |
| walker2d-medium-replay | $10.2 \pm 2.4$ | $11.6 \pm 1.7$ |
| hopper-medium-replay | $29.5 \pm 2.5$ | $30.8 \pm 2.9$ |
| halfcheetah-medium-expert | $75.8 \pm 10.1$ | $71.9 \pm 9.8$ |
| walker2d-medium-expert | $71.2 \pm 22.0$ | $83.3 \pm 31.0$ |
| hopper-medium-expert | $99.9 \pm 29.0$ | $72.1 \pm 20.6$ |
| pen-human | $45.5 \pm 24.5$ | $40.7 \pm 29.5$ |
| pen-cloned | $18.0 \pm 14.4$ | $27.7 \pm 15.1$ |
| pen-expert | $141.1 \pm 14.8$ | $130.8 \pm 11.3$ |
| door-expert | $103.4 \pm 3.7$ | $104.6 \pm 0.9$ |
| Average Score | **61.4** | 58.8 |

Table 7: Normalized returns under several values of $\sigma \triangleq \sigma_B = \sigma_J$ (Appendix D.2.1) in the full algorithm of GAN-joint-matching. The reported number are the means and standard deviations of the normalized returns of the last five rollouts across three random seeds $\{0, 1, 2\}$.

| Task Name | $\sigma = 1 \times 10^{-2}$ | $\sigma = 3 \times 10^{-3}$ | $\sigma = 1 \times 10^{-3}$ | $\sigma = 3 \times 10^{-4}$ | $\sigma = 1 \times 10^{-4}$ | $\sigma = 0$ |
|---|---|---|---|---|---|---|
| maze2d-umaze | $48.4 \pm 21.4$ | $48.3 \pm 19.7$ | $41.7 \pm 11.5$ | $40.1 \pm 16.9$ | $54.9 \pm 10.4$ | $50.8 \pm 24.0$ |
| maze2d-medium | $58.7 \pm 33.6$ | $48.7 \pm 7.4$ | $64.0 \pm 23.9$ | $69.6 \pm 25.6$ | $46.9 \pm 15.5$ | $26.4 \pm 5.7$ |
| maze2d-large | $87.1 \pm 17.9$ | $57.6 \pm 21.3$ | $62.4 \pm 13.3$ | $71.3 \pm 26.0$ | $61.0 \pm 8.6$ | $62.3 \pm 32.3$ |
| halfcheetah-medium | $43.0 \pm 0.4$ | $43.7 \pm 0.3$ | $43.9 \pm 0.4$ | $44.1 \pm 0.3$ | $43.8 \pm 0.3$ | $44.0 \pm 0.4$ |
| walker2d-medium | $56.9 \pm 10.4$ | $66.4 \pm 7.9$ | $68.8 \pm 10.3$ | $69.3 \pm 8.6$ | $64.6 \pm 13.8$ | $63.8 \pm 8.4$ |
| hopper-medium | $23.5 \pm 8.4$ | $66.7 \pm 20.8$ | $63.3 \pm 21.0$ | $60.1 \pm 27.3$ | $74.5 \pm 19.3$ | $89.6 \pm 27.9$ |
| halfcheetah-medium-replay | $32.1 \pm 2.4$ | $31.5 \pm 3.3$ | $32.3 \pm 2.1$ | $33.1 \pm 2.3$ | $31.2 \pm 1.9$ | $31.5 \pm 3.2$ |
| walker2d-medium-replay | $9.8 \pm 2.4$ | $10.7 \pm 2.0$ | $10.2 \pm 1.8$ | $10.2 \pm 2.4$ | $10.9 \pm 1.7$ | $9.4 \pm 1.4$ |
| hopper-medium-replay | $28.7 \pm 3.8$ | $30.1 \pm 2.7$ | $30.5 \pm 2.9$ | $29.5 \pm 2.5$ | $31.3 \pm 1.9$ | $29.2 \pm 1.5$ |
| halfcheetah-medium-expert | $79.9 \pm 10.1$ | $74.2 \pm 13.0$ | $76.8 \pm 13.4$ | $75.8 \pm 10.1$ | $71.3 \pm 8.6$ | $70.7 \pm 7.9$ |
| walker2d-medium-expert | $67.4 \pm 16.0$ | $63.4 \pm 22.2$ | $69.7 \pm 17.3$ | $71.2 \pm 22.0$ | $63.4 \pm 23.1$ | $77.2 \pm 18.4$ |
| hopper-medium-expert | $20.5 \pm 6.8$ | $56.7 \pm 27.5$ | $79.4 \pm 21.9$ | $99.9 \pm 29.0$ | $66.7 \pm 19.6$ | $62.0 \pm 19.5$ |
| pen-human | $-3.3 \pm 0.5$ | $64.2 \pm 17.0$ | $46.6 \pm 33.5$ | $45.5 \pm 24.5$ | $67.8 \pm 13.4$ | $60.3 \pm 11.4$ |
| pen-cloned | $4.5 \pm 1.9$ | $19.6 \pm 11.8$ | $23.3 \pm 13.2$ | $18.0 \pm 14.4$ | $36.6 \pm 18.4$ | $40.0 \pm 20.8$ |
| pen-expert | $74.2 \pm 26.6$ | $132.8 \pm 11.1$ | $132.8 \pm 17.9$ | $141.1 \pm 14.8$ | $136.6 \pm 10.8$ | $132.0 \pm 19.4$ |
| door-expert | $29.1 \pm 9.7$ | $104.1 \pm 1.6$ | $104.2 \pm 1.7$ | $103.4 \pm 3.7$ | $102.9 \pm 3.9$ | $102.3 \pm 4.8$ |
| Average Score | $41.3$ | $57.4$ | $59.4$ | $61.4$ | $60.3$ | $59.5$ |

## F    FURTHER COMPARISON BETWEEN IMPLICIT AND GAUSSIAN POLICY

In Section 5.2 we note that a uni-modal stochastic policy, such as the Gaussian policy, is less flexible to capture all the rewarding actions, on which an implicit policy may fit well. In this section we visualize such a difference.

Figure 2 compares the fitting of the eight-Gaussian toy dataset by implicit policy and Gaussian policy. Specifically, Figure 2a plots the dataset; Figure 2b plots CGAN with the default implicit generator (implicit policy) fitted by the conditional-distribution matching approach; Figure 2c plots CGAN with Gaussian generator (Gaussian policy) fitted by the conditional-distribution matching approach; Figure 2d plots CGAN with implicit policy fitted by the basic joint-visitation matching strategy (Section 4.1); Figure 2e plots CGAN with Gaussian policy fitted by the basic joint-visitation matching strategy. Experimental details are on Appendix D.1.



|     (a) Truth     |     (b) CGAN     |     (c) G-CGAN     |     (d) GAN     |     (e) G-GAN     |

Figure 2: Performance of approximating the behavior policy on the eight-Gaussian dataset by conditional GAN with default (implicit) generator and Gaussian generator. A conditional GAN ("CGAN") and a Gaussian-generator conditional GAN ("G-CGAN") are fitted using the conditional-distribution matching approach. A conditional GAN ("GAN") and a Gaussian-generator conditional GAN ("G-GAN") are fitted using the basic joint-visitation matching strategy (Section 4.1). Performance is judged by **(1)** clear concentration on the eight centers, and **(2)** smooth interpolation between centers, which implies a good and smooth fit to the behavior policy. Details of this toy experiment are presented on Appendix D.1.

We see that whatever training strategies, Gaussian policies fail to learn multi-modal state-conditional action distributions, even if needed. Even though the Gaussian policy version of CGAN may still correctly capture some modes in the action distributions, an improvement over the mode-covering CVAE, they miss other modes. Besides, these Gaussian policy versions interpolate less-smoothly between the centers. In offline RL, these weaknesses is related the missing of some rewarding actions and less-predictable action-choices at unseen states.

To visualize the differences between the implicit and the Gaussian policy in the offline RL setting, we plot the kernel density estimates of the distributions of the first two action-dimensions in the "maze2d-umaze-v1" dataset, where a performance difference is shown in Table 4. Specifically, Figure 3a plots the distribution of actions in the offline dataset. Figure 3b and 3c respectively plot action distributions produced by the final Gaussian policy and the final implicit policy generating Table 4.

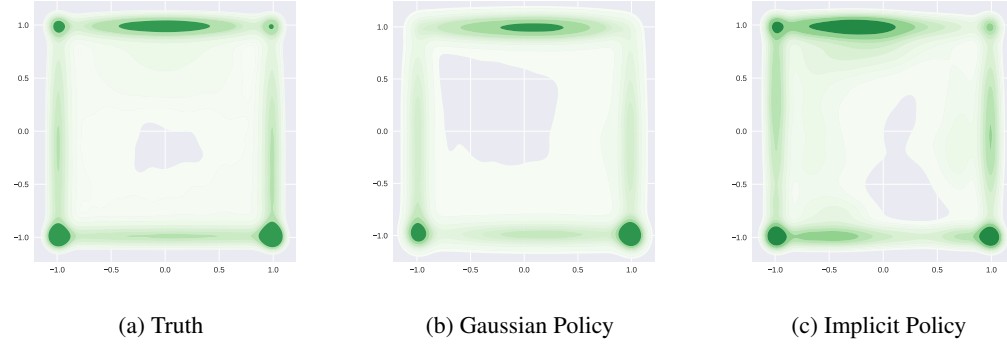

|     (a) Truth     |     (b) Gaussian Policy     |     (c) Implicit Policy     |

Figure 3: KDE plots of the first two dimensions of actions in the "maze2d-umaze-v1" dataset. From left to right: **(a)** Action distribution in the offline dataset; **(b)** Action distribution produced by the final Gaussian policy in Table 4; **(c)** Action distribution by the final implicit policy (Section 4.1) in Table 4.

We see from Figure 3a and Figure 3b that Gaussian policy leaves out two action modes, namely, modes on the upper-left and upper-right corners. Figure 3c shows that our implicit policy does capture all the modes shown in Figure 3a. Note that "maze2d-umaze" is a navigation task requiring agents to reach a goal location (Fu et al., 2021). Gaussian policy thus may miss out some directions in the offline dataset pertaining to short paths to the goal state, which may explain its inferior performance on this dataset in Table 4.

