# OpenReview forum: "State-Action Joint Regularized Implicit Policy for Offline Reinforcement Learning"
_ICLR.cc/2022/Conference — ICLR 2022 Submitted_

### Official Review · Reviewer_aVY7 · 2021-11-01

**Correctness:** 3
**Technical Novelty And Significance:** 3
**Empirical Novelty And Significance:** 3
**Recommendation:** 6
**Confidence:** 4

**Main Review:**

$\textbf{Strengths:}$
&nbsp;

Method
- This direct matching state-action joint access alleviates the problem of uncontrollable extrapolation errors in the estimation of the Q-value function, because the proximity of the access frequency leads to a decrease in the chance of estimating the Q-value of the state-action pair far away from the offline data set.
- Since the state-action pairs in the offline data set are all regarded as samples from joint visits, the problem in the policy matching method that fits the distribution of each state-conditional action on only one data point can be avoided.
- This method implicitly encourages the smoothness of state-action mapping, that is, similar states should have similar actions. This smoothing feature helps to ensure that the policy is reliable when promoting to unseen states.

Experiments
- The authors show the effectiveness of implicit policy, state-action-visitation matching, and their full algorithm in the experimental results. I believe the most valuable experiments are the toy experiments to illustrate their motivations as the findings in the toy experiments may help the policy constraint methods that need to recover the behavior policies to improve their methods further.

Writing
- The paper is well structured and clearly written. I enjoy reading it.

&nbsp;

$\textbf{Weaknesses:}$
&nbsp;

Experiments
- The experimental results seem not very prominent. One reason may be because the experimental setting does not highlight the proposed method’s advantages as there are too few modalities in the used D4RL offline data. I suggest the authors to verify their method on datasets with more modalities to further discover the method’s full potential.



**Summary Of The Paper:**

One of the existing Offline RL algorithms is to constrain the learned policy, such as constraining the learned policy to be consistent with the behavior policy itself or the action distribution based on state conditions, or adopting a Gaussian policy. However, in any given state s, the potential action value function in the action space may have multiple local maxima. Therefore, only one point can be used to evaluate whether the current policy is close to the behavior policy in any specific state, which may not well reflect the true difference between the two conditional distributions, and the stochastic policy that leads to determinism or unimodality may only capture one of the local optima leads to ignoring many other high-value actions. Especially when such policies such as CVAE and other models exhibit strong model-recovering behavior, large probability density may be assigned to low data density areas, resulting in exaggerating the density of high-value actions.

In response to the above problems, the method proposed in this article:
- Uses an implicit policy to capture the multi-modality of the action value function
- Controls the <s,a> access frequency in the data set to be as consistent as possible with the <s,a> access frequency of the learned policy as an additional training target, without the need to explicitly construct a behavior policy
- provides the theory that proves, the method of matching behavior policy and current policy is equivalent to matching their corresponding <s,a> access frequency


**Summary Of The Review:**

This paper developed a framework that supports learning a flexible while well-regularized policy in offline RL. The technical novelty is sufficient and significant. The empirical results show the method reach satisfied performance on D4RL while I'm looking forward to seeing potentially better results on datasets with more modalities. Overall, I recommend to accept this paper.

---

> ### Author Response · Authors · 2021-11-15
> **Response to Reviewer aVY7**
>
> **1**
> > The experimental results seem not very prominent. One reason may be because the experimental setting does not highlight the proposed method’s advantages as there are too few modalities in the used D4RL offline data. I suggest the authors to verify their method on datasets with more modalities to further discover the method’s full potential.
>
> Thanks for the suggestion. We show in Figure 3, Appendix F (page 26 - 27), that the Gaussian policy may leave out action modes in the "maze2d-umaze-v1" dataset that we test on, while our implicit policy does capture all the modes. Based on our visualization and the description of this dataset in [1], we think that multiple action modes are possible and necessary in this dataset. We notice that the performance of our implicit policy on this dataset is better, relatively significantly, than its Gaussian policy counterpart and prior work that adopts the Gaussian policy, as shown in Table 4 (page 24) and Table 1 (page 9), respectively. These visualizations and numerical results, together with our ablation study in Section 5.2 (a), may support our conclusion that the flexibility of our implicit policy helps in capturing multiple action modes in the offline dataset, and hence improves the performance.
>
> We note that apart from the benefits of our implicit policy, as discussed in Section 4.1 and as mentioned in your review, our choice of directly matching the state-action joint visitations circumvents the problem in the prior policy-regularization approach of fitting each state-conditional action distributions on only one data point. Our algorithmic design and enhancement technique of state-smoothing developed in Section 4.2 encourage the smoothness of the state-action mapping, which helps a reliable, predictable, and smooth behavior at states unseen in the offline dataset. We believe that a wide range of offline-RL datasets can benefit from these features, apart from datasets with more modalities, which is confirmed by our experimental results.
>
> Finally, we note that in order to advocate a hyperparameter agnostic setting, we try to minimize hyperparameter tuning across datasets in our experiments, as shown in Table 2 (page 22). Due to the diverse nature of the tested datasets, the distinction between our proposed algorithm and the baselines can be more significant should one be allowed to per-dataset tune the hyperparameters.
>
> [1] Justin Fu, et al. “D4RL: Datasets for Deep Data-Driven Reinforcement Learning.” In arXiv:2004.07219.

---

### Official Review · Reviewer_Ht1b · 2021-11-01

**Correctness:** 2
**Technical Novelty And Significance:** 2
**Empirical Novelty And Significance:** 3
**Recommendation:** 3
**Confidence:** 4

**Main Review:**

Strengths
1.	The paper is well written and easy to follow. It provides good motivation for using implicit policies and regularizing the state-action visitation distribution. The visualization in Figure 1 is also helpful.

Weaknesses
1.	The theoretical analysis assumes $T_{i,j} > 0$. Although the paper claims that the assumption can be satisfied by substituting the zero entries with small numbers, doesn’t that change $||d_\phi – d_b||_1$? Does that bound still hold (how much error would be introduced if we substitute all zero entries)?
2.	I don’t understand the purpose of the theoretical analysis. The paper mentions that the goal is to show that “this line of prior work is compatible in theory with our proposed framework”, but it seems like the theory just says that regularizing policy implies regularizing the state-action visitation. Is that what the paper trying to show or do I miss anything here? Moreover,  I think similar analysis has been done in the existing works (e.g., Appendix A and B in [1] or Section 4.2 of [2]). Can you explain how the analysis in the paper differs from the existing works? What is novelty in the theoretical analysis in this paper?
3.	In the empirical section, the paper mentions that “we do not test on the random and expert datasets as they are less practical”. However, I think the point of the empirical study is to understand how the proposed algorithm performs compared to baselines, not how practical the algorithm or the dataset is. Therefore, I think the empirical results would be clearer with all dataset reported.
4.	The results in Table 1 do not show confidence interval for baselines and the average scores. I find it hard to see a clear conclusion from the table. For example, for many tasks, it is not clear whether the proposed algorithm outperforms baselines significantly or not. Moreover, showing the proposed algorithms have higher average scores seems pointless since the scores are not normalized (please correct me if I am wrong). It is possible that one algorithm wins on one task with a big margin and loses on other tasks with a small margin to have a higher average score. Is this algorithm better?
5.	One of the main points of the paper is about regularizing the state-action visitation distribution.  However, in Section 4.1, the paper uses the state distribution from the dataset to approximate $d_{\phi}(s)$. Doesn’t the algorithm basically reduce to policy regularizing algorithm by doing this approximation? Approximating the state-action visitation of a target policy is still an ongoing topic in the OPE community, so I don’t think it is okay to just say that we can use the dataset to approximate the state distribution. This is my biggest concern of the paper.

[1] John  Schulman, et al. “Trust region policy optimization.”
[2] Sergey Levine, et al. “Offline Reinforcement Learning: Tutorial, Review, and Perspectives on Open Problems”


**Summary Of The Paper:**

The paper proposes offline RL algorithms which consist of several key components: (1) use implicit policies instead of Gaussian policies (2) regularize the state-action visitation distribution instead of the policy. The paper provides a theoretical analysis to show the equivalence between regularizing policy and regularizing state-action visitation, and provides an empirical study on several deep RL environments.

**Summary Of The Review:**

I am leaning towards a recommendation to reject this paper since the proposed algorithm does not properly regularize the state-action visitation, and the theoretical and empirical results do not support the main points of this paper as mentioned in the Main Review section.

---
After the author response, I think the paper provides a good empirical study on state-action joint regularization methods, so I increased my score for the empirical novelty and significance. However, my main concern remains so I am keeping my overall score.

---

> ### Author Response · Authors · 2021-11-15
> **Response to Reviewer Ht1b (Part 1)**
>
> **1**
> > The theoretical analysis assumes $T_{i,j} > 0$. Although the paper claims that the assumption can be satisfied by substituting the zero entries with small numbers, doesn’t that change $||d_\phi - d_b||_1$? Does that bound still hold (how much error would be introduced if we substitute all zero entries)?
>
> The assumption of $T_{i,j} > 0$ is a technical condition conveniently chosen for easy analysis, which mathematically ensures that $1$ is an eigenvalue of $T^{\top}$ with algebraic and geometric multiplicity $1$. We think that this assumption is reasonable in practice, since the opposite means that some states have absolutely zero chance of reaching some other states, which may be restrictive.  Informally speaking, since the transition matrix $T$ is stochastic with eigenvalues bounded by $1$ and the state visitation $d$ satisfy the property that $(I - T^{\top}) d = 0$, it follows that a small perturbation to the zero entries in $T$ will also lead to a small perturbation in $d$. Since the perturbation to $T$ can be make sufficiently small, the bound in Theorem 1 on $||d_\phi - d_b||_1$ should not be significantly affected. We note that a rigorous analysis may need to take into account the number and the distribution of zeros in the transition matrix $T$, which may be complicated and may require further assumptions.
>
> **2.1**
> > I don’t understand the purpose of the theoretical analysis. The paper mentions that the goal is to show that “this line of prior work is compatible in theory with our proposed framework”, but it seems like the theory just says that regularizing policy implies regularizing the state-action visitation. Is that what the paper trying to show or do I miss anything here?
>
> In the theoretical analysis, we show that the prior work in offline RL of controlling the distance between the behavior policy and current policy, in essence, controls their corresponding state-action visitations. This analysis theoretically links a line of prior offline-RL work with our proposed framework, manifesting the generality of our algorithmic idea and the proposed framework.
>
> We note that even though previous policy-regularization algorithms also theoretically match the state-action visitations, in practice, especially in continuous control tasks, they suffer from the problem that each state-conditional action distribution is fitted on only one data point, which may be insufficient to match two conditional distributions. As discussed in Section 4.1, our framework of directly matching the state-action joint visitations circumvents this problem, since state-action pairs in the offline dataset can all be viewed as samples from the joint visitation frequency, instead of each pair being separately viewed as one sample from the corresponding state-conditional action distribution, *i.e.*, $a_i \sim \pi(\cdot | s_i)$.

---

> > ### Author Response · Authors · 2021-11-15
> > **Response to Reviewer Ht1b (Part 2)**
> >
> > **2.2**
> > > Moreover, I think similar analysis has been done in the existing works (e.g., Appendix A and B in [1] or Section 4.2 of [2]). Can you explain how the analysis in the paper differs from the existing works? What is novelty in the theoretical analysis in this paper?
> >
> > We thank the reviewer for pointing out related work.
> >
> > Section 4.2 of [2] gives a bound of $D_{KL}(d^\pi(s) || d^{\pi_{\beta}}(s))$ by $O(\epsilon / (1-\gamma)^2)$, which depends on $1-\gamma$ on the denominator. Hence, this bound cannot be applied to the undiscounted MDP ($\gamma = 1$), and can be very loose when the discount factor $\gamma$ is close to 1, which is typically the case in the default hyperparameter of prior offline-RL algorithms. By contrast, our bound avoids the dependency on the discount factor $\gamma$ and hence can apply equally well on both discounted and undiscounted MDP. In this sense, our theoretical result is more general and has wider applicability. Besides, Section 4.2 of [2] bounds the KL divergence between the state visitation frequencies while we bound their total variation distance, which is a well-defined metric compared to the KL divergence. Furthermore, we also consider the case where the behavior policy in the dataset is a mixture of data-collecting policies, which is not discussed in Section 4.2 of [2].
> >
> > Appendix A and B in [1] are basically the reference of Section 4.2 of [2]. Apart from what has been discussed, we note that Appendix A and B in [1] are devoted to the proof of the policy improvement bound (Theorem 1 of [1]), while our analysis tries to link a line of prior offline-RL work with our proposed framework on a general setting without specific requirements on the underlying MDP, *e.g.*, a discounted MDP ($\gamma < 1$). To the best of our knowledge, such analysis has not been clearly done in the offline RL literature. We agree with the reviewer that we use a similar analysis technique as in Appendix B of [1], the technique of perturbation analysis, which is classical in the literature of numerical linear algebra (e.g. [3] and [4]).
> >
> > [1] John Schulman, et al. “Trust region policy optimization.” In ICML 2015.
> >
> > [2] Sergey Levine, et al. “Offline Reinforcement Learning: Tutorial, Review, and Perspectives on Open Problems.” In arXiv:2005.01643.
> >
> > [3] Lloyd Trefethen and David Bau. “Numerical Linear Algebra.” SIAM 1997.
> >
> > [4] Gene Golub and Charles Van Load. “Matrix Computations.” The Johns Hopkins University Press 2013.

---

> > > ### Author Response · Authors · 2021-11-15
> > > **Response to Reviewer Ht1b (Part 3)**
> > >
> > > **3**
> > > > In the empirical section, the paper mentions that “we do not test on the random and expert datasets as they are less practical”. However, I think the point of the empirical study is to understand how the proposed algorithm performs compared to baselines, not how practical the algorithm or the dataset is. Therefore, I think the empirical results would be clearer with all dataset reported.
> > >
> > > Based on the experiments and conclusions in [1], for sufficiently diverse offline datasets such as the "random" datasets, standard off-policy RL algorithms can work well, which is also shown numerically on Table 2 and Table 3 of [2]. Therefore, it is typically considered as unnecessary to design and test offline RL algorithms on the "random” datasets. [3] notes that "offline RL works ... typically collect the static batch with a *partially trained policy* rather than a random policy.” Hence, the "random” datasets may not be a typical assessment for the offline RL algorithms. Besides, many works in offline RL also do not test on the "random” dataset, such as [4], [5], and [6].
> > >
> > > The "expert” datasets can often be very well solved using behavior cloning algorithms and hence, design and test of offline RL algorithms on these datasets are usually unnecessary. We also note that evaluations on the "expert” datasets are not included in [2] and this dataset currently lacks a comprehensive benchmarking of prior offline-RL algorithms.
> > >
> > > Due to the above reasons and following the previous work, we do not conduct experiments on the "random” and "expert” datasets. Nevertheless, if that is indeed the demand of Reviewer Ht1b, we can add results on the "random” and "expert” datasets.
> > >
> > > We would like to note that apart from the standard Gym-MuJoCo datasets, we further test our algorithms on the Adroit and Maze2D datasets, which respectively represent high-dimensional sparse-reward-signal datasets and datasets collected by non-Markovian policy. [2] points out these datasets can be challenging for existing methods and we believe that testing on these datasets can provide a more comprehensive evaluation of our algorithms. From the experimental results, our algorithms perform robustly and comparatively-well on these datasets, which supports the efficacy of our algorithmic idea and proposed framework.
> > >
> > > [1] Rishabh Agarwal, et al. “An Optimistic Perspective on Offline Reinforcement Learning”. In ICML 2020.
> > >
> > > [2] Justin Fu, et al. “D4RL: Datasets for Deep Data-Driven Reinforcement Learning.” In arXiv:2004.07219.
> > >
> > > [3] Tatsuya Matsushima, et al. “Deployment-Efficient Reinforcement Learning via Model-Based Offline Optimization” In ICLR 2021.
> > >
> > > [4] Catherine Cang, et al. “Behavioral Priors and Dynamics Models: Improving Performance and Domain Transfer in Offline RL.” In arXiv:2106.09119.
> > >
> > > [5] Lili Chen, et al. “Decision Transformer: Reinforcement Learning via Sequence Modeling.” In NeurIPS 2021.
> > >
> > > [6] Ilya Kostrikov, et al. “Offline Reinforcement Learning with Implicit Q-Learning.” In arXiv:2110.06169.
> > >
> > > **4.1**
> > > > The results in Table 1 do not show confidence interval for baselines and the average scores.
> > >
> > > Due to limited computational resources, we use the evaluation results in [1] for the baseline algorithms "SAC-off”, "BEAR”, "BRAC-v” and "BCQ”, where confidence intervals are not included.
> > > Since CQL has multiple hyperparameter recommendations for each domain of tasks, we rerun CQL using all the applicable recommended hyperparameters and per-task per-random-seed pick the best results across the hyperparameter recommendations. Therefore, confidence intervals for the CQL results may not be fairly calculated. For the average scores, we follow the literature (listed in the Response **4.3**) to not calculate the confidence intervals.
> > >
> > > [1] Justin Fu, et al. “D4RL: Datasets for Deep Data-Driven Reinforcement Learning.” In arXiv:2004.07219.
> > >
> > > **4.2**
> > > > I find it hard to see a clear conclusion from the table. For example, for many tasks, it is not clear whether the proposed algorithm outperforms baselines significantly or not.
> > >
> > > As discussed in the following response (Response **4.3**), the reported scores in Table 1 has been *normalized*, and hence the reported average *normalized* scores is commonly used as an assessment of the general performance of the algorithms. We add the **Average Rank** of each algorithm in this revision, which provides a robust assessment of the general performance of each algorithm.
> > >
> > > We note that in order to advocate a hyperparameter agnostic setting, we try to minimize hyperparameter tuning across datasets in our experiments, as shown in Table 2 (page 22). Due to the diverse nature of the tested datasets, the distinction between our proposed algorithm and the baselines can be more significant should one be allowed to per-dataset tune the hyperparameters.

---

> > > > ### Author Response · Authors · 2021-11-15
> > > > **Response to Reviewer Ht1b (Part 4)**
> > > >
> > > > **4.3**
> > > > > Moreover, showing the proposed algorithms have higher average scores seems pointless since the scores are not normalized (please correct me if I am wrong).
> > > >
> > > > In Table 1, where the average scores are shown, the reported score in each task has been normalized, as stated in the caption of Table 1. The normalization is performed according to the evaluation protocol provided by the D4RL package [1], whose purpose is “to facilitate comparison across tasks’’.
> > > >
> > > > Average *normalized* scores across the tested tasks has been used in the offline RL literature to assess and compare the performance of algorithms, for example, in Section 6 of [1], Table 2 of [2], Table 1 of [3], Table 2 of [4], and Table 1 of [5]. Therefore, showing our proposed algorithms have higher average *normalized* scores in Table 1 is a fair assessment.
> > > >
> > > > [1] Justin Fu, et al. “D4RL: Datasets for Deep Data-Driven Reinforcement Learning.” In arXiv:2004.07219.
> > > >
> > > > [2] Rahul Kidambi, et al. “MOReL: Model-Based Offline Reinforcement Learning.” In NeurIPS 2020.
> > > >
> > > > [3] Catherine Cang, et al. “Behavioral Priors and Dynamics Models: Improving Performance and Domain Transfer in Offline RL.” In arXiv:2106.09119.
> > > >
> > > > [4] Lili Chen, et al. “Decision Transformer: Reinforcement Learning via Sequence Modeling.” In NeurIPS 2021.
> > > >
> > > > [5] Samarth Sinha, et al. “S4RL: Surprisingly Simple Self-Supervision for Offline Reinforcement Learning.” In CoRL 2021.
> > > >
> > > > **4.4**
> > > > > It is possible that one algorithm wins on one task with a big margin and loses on other tasks with a small margin to have a higher average score. Is this algorithm better?
> > > >
> > > > Thanks for raising this concern. We agree that a small amount of big-margin wins may distort the Average Score and hide many small-margin losses. To address this, we calculate the *median* of the normalized scores and the average ranking of each algorithm across the tested 16 tasks in the following table, respectively in the row **Median Score** and **Average Rank**, where a high Median Score and a low Average Rank are desirable. Since the median and the ranking are more robust to numerical outliers, they could avoid the mentioned undesirable situation in the result presentation.
> > > >
> > > > | Algorithm     | SAC-off | BEAR | BRAC-v | BCQ  | CQL  | GAN-Cond:Basic | GAN-Joint:Basic | GAN-Joint |
> > > > |---------------|---------|------|--------|------|------|----------------|-----------------|-----------|
> > > > | Average Score | 10.0    | 44.1 | 24.1   | 51.4 | 47.1 | 55.6           | 59.6            | 61.4      |
> > > > | Median Score  | 1.9     | 39.4 | 16.0   | 48.6 | 41.2 | 52.6           | 59.3            | 64.7      |
> > > > | Average Rank  | 6.8     | 4.8  | 5.5    | 4.3  | 4.6  | 3.9            | 2.9             | 3.2       |
> > > >
> > > > From the above table, it is clear that the median scores of the tested three implementations of our framework are all close to the average scores previously reported, while the evaluation of the baseline algorithms witness drops. The best three algorithms in terms of the Average Rank are again the three implementations of our framework, and our two joint-matching implementations win the others by a margin. This table coincides with our conclusion in Section 5.1 that **(1)** our state-action-visitation joint-matching algorithms overall outperform the baseline algorithms, and **(2)** their performances are relatively stable across tasks and datasets that possess diverse nature.
> > > >
> > > > **5.1**
> > > > > … Doesn’t the algorithm basically reduce to policy regularizing algorithm by doing this approximation?
> > > >
> > > > As shown in [1] and [2], policy regularizing algorithms are implemented by using *the same state* in both the generator sample $x$ and the data sample $y$. In our work, as shown in Equation 6 (page 6) and Equation 9 (page 7), we construct the generator sample $x$ by making a new sample of state independent of the state in $y$ and adding noise to it, which is different from prior policy-regularization algorithms.
> > > >
> > > > In our work, we are motivated by the prior works in offline RL to develop an approximation to $d_\phi(s)$ that is akin to a kernel density approximation with data points $s \in D$ and with a radial basis kernel of bandwidth $\sigma_J$. We will discuss this further in the next response (Response **5.2**).
> > > >
> > > > [1] Aviral Kumar, et al. “Stabilizing Off-Policy Q-Learning via Bootstrapping Error Reduction.” In NeurIPS 2019.
> > > >
> > > > [2] Yifan Wu, et al. “Behavior Regularized Offline Reinforcement Learning.” In arXiv:1911.11361.

---

> > > > > ### Author Response · Authors · 2021-11-15
> > > > > **Response to Reviewer Ht1b (Part 5)**
> > > > >
> > > > > **5.2**
> > > > > > Approximating the state-action visitation of a target policy is still an ongoing topic in the OPE community, so I don’t think it is okay to just say that we can use the dataset to approximate the state distribution. This is my biggest concern of the paper.
> > > > >
> > > > > To the best of our knowledge and as per you mentioned, approximating the state-action visitation of a target policy is still an open problem. In our work, we are motivated by the off-policy and offline policy optimization to use the state visitation of the behavior policy, $d_b(s)$, to approximate that of the current policy, $d_\phi(s)$. Combined with the state-smoothing technique we develop in Section 4.2, our approximation is akin to a kernel density approximation to $d_\phi(s)$ with data points $s \in D$ and with a radial basis kernel of bandwidth $\sigma_J$.
> > > > >
> > > > > Based on the experimental results, our algorithms perform stably and generally-well across a diverse set of tasks. It therefore implies that the issue of out-of-distribution state-action pair is alleviated by our proposed regularization, since policies without proper regularization may fail in typical offline RL datasets, as discussed in [1] and [2]. From the construction of our regularization, our approximation could still effectively regularize the state-action visitation of the current policy to be close to that of the behavior policy during the training process.
> > > > >
> > > > > Note that several offline RL algorithms, such as [2] and [3], also adopt similar approximations in both the policy optimization and the policy regularization. In this work, we are partially motivated by their algorithmic design.
> > > > >
> > > > > Finally, we thank the reviewer for raising this concern. Since our proposed framework is highly modular, advances in the OPE community on this open problem may be easily incorporated into our framework. We will continue tracing this research topic in our future work.
> > > > >
> > > > > [1] Scott Fujimoto, et al. “Off-Policy Deep Reinforcement Learning without Exploration.” In ICML 2019.
> > > > >
> > > > > [2] Aviral Kumar, et al. “Stabilizing Off-Policy Q-Learning via Bootstrapping Error Reduction.” In NeurIPS 2019.
> > > > >
> > > > > [3] Yifan Wu, et al. “Behavior Regularized Offline Reinforcement Learning.” In arXiv:1911.11361.

---

> > > > > > ### Comment · Reviewer_Ht1b · 2021-11-16
> > > > > > **Response to Authors**
> > > > > >
> > > > > > Thank you for the response. I am happy with the response for 2, 3, and 4. For 3, I am not asking the authors to run on the "random” and "expert” datasets. I just think it is not well-justified to remove some dataset because they are not “practical”. The reasons mentioned in the response make more sense to me and I agree there is no need to run on the "random” and "expert” datasets.
> > > > > >
> > > > > > For 5, based on the response, I guess the state smoothing technique is the key component (not just an enhancing component). However, I still don’t see how the approximation with state smoothing can be a kernel density approximation to $d_\phi$. The data is sampled from $d_b$, so shouldn’t the approximation with state smoothing be more like an approximation to $d_b$ instead of $d_\phi$?
> > > > > >
> > > > > > Moreover, I think the approximation can be useless (i.e., we can find an example that the approximation is arbitrarily bad). For example, we can construct a MDP with two Markov chains and a starting state. From the starting state, $a_1$ leads to chain 1 and $a_2$ leads to chain 2. The states in chain 1 are very far (and disjoint) from the states in chain 2. If the behavior policy chooses $a_1$ in the starting state and the target policy chooses $a_2$ in the starting state, the data provide no information about $d_\phi$. The only way to avoid this situation is by regularizing the policy in the starting state to choose $a_1$.
> > > > > >
> > > > > > Therefore, I still think the approximation does not make sense to me, and I don’t think the algorithm can regularize the state-action visitation properly as claimed in the paper.  However, I do think the empirical results with state-smoothing are promising. I encourage the author improve the paper with stronger justification for regularization with the state-smoothing technique.
> > > > > >
> > > > > > For 1, I think I am slightly confused now. I think the state-action visitation is the discounted occupancy measure but it seems like the authors are talking about the stationary distribution of a Markov process. Do I miss something here? I would be surprised if the bounds on the discounted occupancy measure does not depend on the discounting factor.

---

> > > > > > > ### Author Response · Authors · 2021-11-17
> > > > > > > **Response to Reviewer Ht1b (Commented on 16 Nov 2021, Part 1)**
> > > > > > >
> > > > > > > Thanks for the response.
> > > > > > >
> > > > > > > **1**
> > > > > > > > For 5, based on the response, I guess the state smoothing technique is the key component (not just an enhancing component).
> > > > > > >
> > > > > > > While we agree that the proposed state-smoothing technique is important and helps enhancing the performance, we would not consider it as a key component. The key component in this paper is the state-action joint regularization framework.
> > > > > > >
> > > > > > > We note that even without the state-smoothing component, our basic state-action-visitation matching algorithm (“GAN-Joint:Basic” in Table 1, page 9) still performs robustly and comparatively-well across the 16 datasets we tested on. From Table 1, it is clear that “GAN-Joint:Basic” outperforms, by a margin, its policy-matching counterpart (“GAN-Cond:Basic”) and the SOTA policy-matching algorithms BEAR and BRAC, in terms of both Average Score and Average Rank. As discussed in our theoretical analysis, the policy-matching approach in essence controls the corresponding state-action visitations. We therefore infer from these empirical results that our framework can effectively regularize the state-action visitation during the training process, since otherwise, our algorithms would suffer from the out-of-distribution state-action pairs and the empirical performance would be significantly deteriorated.
> > > > > > >
> > > > > > > Response **3** below provides further clarification on the reviewer’s concerns about our proposed framework.
> > > > > > >
> > > > > > > **2**
> > > > > > > > However, I still don’t see how the approximation with state smoothing can be a kernel density approximation to $d_\phi$. The data is sampled from $d_b$, so shouldn’t the approximation with state smoothing be more like an approximation to $d_b$ instead of $d_\phi$.
> > > > > > >
> > > > > > > By saying “Our strategy *is akin to* sampling from a kernel density approximation of $d_\phi(s)$”, we try to explain the potential of our proposed state-smoothing technique as using the added continuity and smoothness to better approximate (the samples from) $d_\phi$ with the offline dataset. Based on our understanding, the continuity and smoothness enabled by the kernels is the principal idea behind the kernel density approximation, which motivates our algorithmic design.
> > > > > > >
> > > > > > > We agree with the reviewer that the proposed state-smoothing technique can also be understood as providing a better and smoother approximation of $d_b$. Based on Response **3** below, such an improved approximation of $d_b$ can provide more information about $d_\phi$. We appreciate the reviewer for providing a new viewpoint to our state-smoothing technique.
> > > > > > >
> > > > > > > We encourage the readers to take this as an informal analogy for explaining our proposed enhancing component, not as a rigorous argument.

---

> > > > > > > > ### Author Response · Authors · 2021-11-17
> > > > > > > > **Response to Reviewer Ht1b (Commented on 16 Nov 2021, Part 2)**
> > > > > > > >
> > > > > > > > **3**
> > > > > > > > > Moreover, I think the approximation can be useless (i.e., we can find an example that the approximation is arbitrarily bad). For example, we can construct a MDP with two Markov chains and a starting state. ... The states in chain 1 are very far (and disjoint) from the states in chain 2. If the behavior policy chooses $a_1$ in the starting state and the target policy chooses $a_2$ in the starting state, the data provide no information about $d_\phi$. …
> > > > > > > >
> > > > > > > > By saying “The states in chain 1 are very far (and disjoint) from the states in chain 2.”, the provided example is a reducible Markov chain, where it is *impossible* to go from every state (states in chain 1) to every state (states in chain 2) in the state space. We appreciate the reviewer for raising this insightful concern. However, we would like to note that this concern is currently out of the scope of this paper and the offline-RL and OPE community. Specifically, in the current offline-RL and OPE work, one classical assumption is the ergodicity assumption (*e.g.*, in [1], [2] and [3]), which, as stated in the assumption of our theoretical analysis (Appendix C, page 16-17), assumes that the Markov chains associated with any behavior policy $\pi_b$ and any $\pi_\phi$ under consideration are ergodic.
> > > > > > > >
> > > > > > > > Based on this assumption, the Markov chains on the state space $S$ induced by $\pi_b$ and $\pi_\phi$ are irreducible, which, when applied to your example, means that chain 1 will eventually visit states in chain 2 and *vice versa*. Because of the irreducibility condition, even though the behavior policy chooses $a_1$ and the target policy chooses $a_2$ in the starting state, the data in chain 1 should still provide information about chain 2 and hence $d_\phi$.
> > > > > > > >
> > > > > > > > In general, the ergodicity assumption ensures that in theory, the support of $d_b(s)$ induced by $\pi_b$ should have a non-empty intersection with the support of $d_\phi(s)$ induced by $\pi_\phi$. Otherwise, it is impossible to go from states in the support of $d_b(s)$ to states in the support of $d_\phi(s)$. This means that the Markov chain induced by $\pi_b$ cannot visit states in the support of $d_\phi(s)$, which violates our assumption that the Markov chain on the state space $S$ induced by $\pi_b$ is irreducible. Therefore, because of the ergodicity assumption, data from $d_b$ or the trajectories induced by $\pi_b$ should provide information about $d_\phi$ in theory, which underpins the approximation we used in the paper.
> > > > > > > >
> > > > > > > > We note that this approximation is basically the same approximation used in the offline policy optimization, which is discussed in Section 2 (page 3) and in [4], and has been demonstrated as effective in offline policy optimization. We agree that examples on which our algorithm can fail might exist, however, we have not found the significance of those examples in our experiments.
> > > > > > > >
> > > > > > > > **4**
> > > > > > > > >For 1, I think I am slightly confused now. I think the state-action visitation is the discounted occupancy measure but it seems like the authors are talking about the stationary distribution of a Markov process. Do I miss something here? I would be surprised if the bounds on the discounted occupancy measure does not depend on the discounting factor.
> > > > > > > >
> > > > > > > > We'd like to clarify that while the discounted occupancy measure is widely used in the policy optimization objectives, the bounds in our paper are not on discounted occupancy measure as neither the data collection (i.e., policy rollout of state-action pairs) nor the state-action joint regularization involve the discount factor.
> > > > > > > >
> > > > > > > > [1] Qiang Liu, et al. “Breaking the Curse of Horizon: Infinite-Horizon Off-Policy Estimation.” In NeurIPS 2018.
> > > > > > > >
> > > > > > > > [2] Ofir Nachum, et al. “AlgaeDICE: Policy Gradient from Arbitrary Experience.” In arXiv:1912.02074.
> > > > > > > >
> > > > > > > > [3] Nathan Kallus and Angela Zhou. “Confounding-Robust Policy Evaluation in Infinite-Horizon Reinforcement Learning.” In NeurIPS 2020.
> > > > > > > >
> > > > > > > > [4] Sergey Levine, et al. “Offline Reinforcement Learning: Tutorial, Review, and Perspectives on Open Problems.” In arXiv:2005.01643.

---

> > > > > > > > > ### Comment · Reviewer_Ht1b · 2021-11-17
> > > > > > > > > **Quick Response**
> > > > > > > > >
> > > > > > > > > “We encourage the readers to take this as an informal analogy for explaining our proposed enhancing component, not as a rigorous argument.”
> > > > > > > > > In my opinion, adding informal and weak argument does not improve the paper at all. It could even raise many concerns for the readers. For example, the continuity and smoothness could help, however, what assumption do we need to make it helpful? We probability need to assume the stationary distribution is smooth, but does it make sense to make such assumption?
> > > > > > > > >
> > > > > > > > > “By saying “The states in chain 1 are very far (and disjoint) from the states in chain 2.”, the provided example is a reducible Markov chain”
> > > > > > > > > I think we can easily modify the example to make it irreducible. For example, just adding a small probability for a state in one chain to transit to a state in another chain. We still suffer from a similar issue: we have most data in chain 1 and very little data in chain 2. Simply using the data (e.g., without density ratio correction) does not provide enough information about $d_\phi$.
> > > > > > > > >
> > > > > > > > > “the bounds in our paper are not on discounted occupancy measure as neither the data collection (i.e., policy rollout of state-action pairs) nor the state-action joint regularization involve the discount factor.”
> > > > > > > > > Does it mean the data is sampled from the stationary distribution of the Markov process induced by $\pi_b$?
> > > > > > > > >
> > > > > > > > > Finally, I want to clarify that I am not against the core idea in the paper: state-action joint regularization. In fact, I think the idea is neat (simple and effective). However, I think the regularization has nothing to do with regularization the state-action visitation, which is one of the main claims in the paper. In my opinion, I would view it as an improvement to policy regularization where we sample two states independently instead of conditioning on the same state. I think the paper should not make claims that cannot be supported in the paper.

---

> > > > > > > > > > ### Author Response · Authors · 2021-11-18
> > > > > > > > > > **Response to Reviewer Ht1b (Commented on 17 Nov 2021, Part 1)**
> > > > > > > > > >
> > > > > > > > > > **1**
> > > > > > > > > > > ... For example, the continuity and smoothness could help, however, what assumption do we need to make it helpful? We probability need to assume the stationary distribution is smooth, but does it make sense to make such assumption?
> > > > > > > > > >
> > > > > > > > > > For any density/function estimation problems, such as kernel density estimation given empirical data samples, it is common to make certain assumptions about the smoothness of the underlying distribution/function/decision boundary. Without an appropriate smoothness assumption, one could easily overfit the observed data samples.
> > > > > > > > > >
> > > > > > > > > >
> > > > > > > > > > **2**
> > > > > > > > > > > I think we can easily modify the example to make it irreducible. For example, just adding a small probability for a state in one chain to transit to a state in another chain. We still suffer from a similar issue: we have most data in chain 1 and very little data in chain 2. Simply using the data (e.g., without density ratio correction) does not provide enough information about $d_\phi$.
> > > > > > > > > >
> > > > > > > > > >
> > > > > > > > > > We’d like to clarify that when the data are only from or dominated by chain 1, the proposed state-action joint regularization will push our policy towards chain 1.
> > > > > > > > > >
> > > > > > > > > > We summarize the concern as dealing with small datasets that have narrow data distribution, which is challenging to current offline RL algorithms [1]. We note that the “pen-human” dataset we test on is an example of such datasets [1]. From the experimental results, we see that our algorithms perform well on this dataset, while classical policy-matching algorithms, such as BEAR and BRAC, almost fail.
> > > > > > > > > >
> > > > > > > > > > Last but not least, we would appreciate it if you could provide additional datasets that can be used to challenge our algorithm. As discussed in our previous response, we agree that examples on which our algorithm can fail might exist, however, we have not yet found the existence of those examples in our experiments.
> > > > > > > > > >
> > > > > > > > > > [1] Justin Fu, et al. “D4RL: Datasets for Deep Data-Driven Reinforcement Learning.” In arXiv:2004.07219.
> > > > > > > > > >
> > > > > > > > > > **3**
> > > > > > > > > > > Does it mean the data is sampled from the stationary distribution of the Markov process induced by $\pi_b$?
> > > > > > > > > >
> > > > > > > > > > To the best of our knowledge, offline-RL datasets are typically collected by policy rollouts, which can be viewed as sampling from the undiscounted state-action visitation frequency of the Markov process induced by $\pi_b$. The (undiscounted) state-action visitation frequency equals the stationary distribution induced by $\pi_b$ under the ergodicity assumption discussed in the previous response [1].
> > > > > > > > > >
> > > > > > > > > > [1] Rick Durrett. “Probability: Theory and Examples, Chapter 5.”

---

> > > > > > > > > > > ### Author Response · Authors · 2021-11-18
> > > > > > > > > > > **Response to Reviewer Ht1b (Commented on 17 Nov 2021, Part 2)**
> > > > > > > > > > >
> > > > > > > > > > > **4**
> > > > > > > > > > > > … However, I think the regularization has nothing to do with regularization the state-action visitation, which is one of the main claims in the paper. In my opinion, I would view it as an improvement to policy regularization where we sample two states independently instead of conditioning on the same state. ...
> > > > > > > > > > >
> > > > > > > > > > > We thank the reviewer for agreeing with the core idea of our paper.
> > > > > > > > > > >
> > > > > > > > > > > As a further clarification, the logical flow of the design of our algorithms as as follows;
> > > > > > > > > > > 1. We would like to control the state-action visitation frequency of the current policy $d_\phi(s,a) = d_\phi(s) \pi_\phi(a | s)$ during the training process;
> > > > > > > > > > > 2. Since directly sampling from $d_\phi(s)$ is infeasible in offline RL, we are motivated by the offline policy optimization to use $d_b(s)$ to approximate $d_\phi(s)$.
> > > > > > > > > > >
> > > > > > > > > > > We note that since the offline policy optimization has been shown to be effective in literature, the design choice of using $d_b(s)$ to approximate $d_\phi(s)$ can be effective. Hence, our regularization can approximately control the state-action visitation of the current policy.
> > > > > > > > > > >
> > > > > > > > > > > As we discussed in the previous response, the choice of using $d_b(s)$ to approximate $d_\phi(s)$ is rather an implementation choice. The framework we proposed is highly modular, and may easily incorporate advances in the OPE community on the approximation of $d_\phi(s)$.
> > > > > > > > > > >
> > > > > > > > > > > In general, suppose we have random variables $x, y$ and $x’, y’$, even if $x, x’ \sim p_X(x)$, $y \sim \pi(y | x)$ and $y’$ from a parametric distribution family, we could still regularize on the joint-distribution between $(x, y)$ and $(x’, y’)$. This is a joint matching, since the regularization needs to consider the distributional uncertainty in $y$ across the whole $p_X(x)$, not just one single $x$.
> > > > > > > > > > >
> > > > > > > > > > > From another perspective, mathematically we have $E_{(x,y) \sim p_{X,Y}, (x',y') \sim p'_X \cdot \pi_\phi(y'|x')}  [f(x,y,x',y')] =$
> > > > > > > > > > >
> > > > > > > > > > > $ E_{(x,y) \sim p_{X,Y}, (x',y') \sim p_X\cdot\pi_\phi(y'|x')} \left[\frac{p'_X(x')}{p_X(x')} f(x,y,x',y')\right]$. In our paper, we choose the approximation $\frac{p’_X(x’) }{p_X(x’)}=1$ for simplicity. To the best of our knowledge, accurately and efficiently estimating such a density ratio corrector is currently an open problem in the OPE community, which is out of the scope of this paper.
> > > > > > > > > > >
> > > > > > > > > > > As discussed previously, policy-regularization algorithms regularizes the conditional action distribution between $\pi_\phi(\cdot | s)$ and $\pi_b(\cdot | s)$. By definition and based on the prior works, one need to use *the same state* in both the generator sample and the data sample, *i.e.*, conditioning on the same state. Since “we sample two states independently”, our algorithms deal with the state-action joint visitation, which is different from policy regularization.
> > > > > > > > > > >
> > > > > > > > > > > Hope this response and the previous responses about the ergodicity assumption can be helpful to the reviewer.

---

> > > > > > > > > > > > ### Comment · Reviewer_Ht1b · 2021-11-21
> > > > > > > > > > > > **Response to Authors**
> > > > > > > > > > > >
> > > > > > > > > > > > **2**
> > > > > > > > > > > > "the proposed state-action joint regularization will push our policy towards chain 1.”
> > > > > > > > > > > >
> > > > > > > > > > > > Right, but isn’t this mainly due to the policy regularization at the starting state to force the policy to choose $a_1$?
> > > > > > > > > > > >
> > > > > > > > > > > > "From the experimental results, we see that our algorithms perform well on this dataset, while classical policy-matching algorithms, such as BEAR and BRAC, almost fail.”
> > > > > > > > > > > >
> > > > > > > > > > > > I would like to clarify that my main concern isn’t about the empirical performance of the proposed method. My main concern is on the technical correctness. I don’t agree with the paper claiming that the method is regularizing the state-action visitation since the approximation can be arbitrarily loose. For me, it is like claiming that one can approximately do OPE with all IS ratios = 1.
> > > > > > > > > > > >
> > > > > > > > > > > > **3**
> > > > > > > > > > > > I am not sure what is the undiscounted state-action visitation since if you sum up to infinite (for infinite horizon case), the visitation would be infinite unless you divide by the time step (e.g., the average visitation in [1]).
> > > > > > > > > > > >
> > > > > > > > > > > > I don’t see how the probability distribution induced from policy rollouts equals to the stationary distribution. To sample from the stationary distribution, I think you need rollout the policy for a sufficient number of steps (e.g., mixing time) and collect the data afterwards.
> > > > > > > > > > > >
> > > > > > > > > > > > **4**
> > > > > > > > > > > > "policy-regularization algorithms regularizes the conditional action distribution”
> > > > > > > > > > > >
> > > > > > > > > > > > I would argue this is one of many policy-regularization approaches and one can also do state-action joint policy regularization (as used in this paper). Essentially, both approaches are regularizing the policy to choose actions closer to the behavior policy.
> > > > > > > > > > > >
> > > > > > > > > > > > [1] Qiang Liu, et al. “Breaking the Curse of Horizon: Infinite-Horizon Off-Policy Estimation.” In NeurIPS 2018.

---

> > > > > > > > > > > > > ### Author Response · Authors · 2021-11-21
> > > > > > > > > > > > > **Response to Reviewer Ht1b (Commented on 21 Nov 2021)**
> > > > > > > > > > > > >
> > > > > > > > > > > > > **1**
> > > > > > > > > > > > > > Right, but isn’t this mainly due to the policy regularization at the starting state to force the policy to choose $a_1$ ?
> > > > > > > > > > > > >
> > > > > > > > > > > > > As discussed in our paper, our viewpoint of state-action-visitation joint-matching is more general than the policy regularization, so we do not consider it a strict dichotomy the effect of state-action-visitation joint-matching and the effect of policy regularization.
> > > > > > > > > > > > >
> > > > > > > > > > > > > In fact, when our algorithms are matching the state-action joint visitations, the current policies should also be regularized towards the behavior policy to some extent, since very different policies should not have similar state-action joint visitations.
> > > > > > > > > > > > >
> > > > > > > > > > > > > **2**
> > > > > > > > > > > > > > I don’t agree with the paper claiming that the method is regularizing the state-action visitation since the approximation can be arbitrarily loose. For me, it is like claiming that one can approximately do OPE with all IS ratios = 1.
> > > > > > > > > > > > >
> > > > > > > > > > > > > As stated in many of the previous responses, we note that:
> > > > > > > > > > > > > 1. Examples where “the approximation can be arbitrarily loose” might exist, however, we have not yet found the existence of those examples in the standard offline-RL benchmark datasets that we test on.
> > > > > > > > > > > > > 2. Since this paper focuses on offline RL, addressing this on-going topic in OPE is out of the scope of our paper. Besides, [2] notes that applying current OPE methods in the high-dimensional state and action spaces still faces major challenges. Hence, we choose to not use OPE methods but make this simple approximation, motivated by the literature. Meanwhile, the framework we proposed is highly modular, and may easily incorporate advances in the OPE community.
> > > > > > > > > > > > > 3. [2] notes that current OPE estimators “are most suitable in the case where the policy only deviates by a limited amount from the behavior policy.” In such a case where current OPE methods may be applicable, we believe our such approximation is sensible and it may not be compulsory to apply the OPE techniques.
> > > > > > > > > > > > >
> > > > > > > > > > > > >
> > > > > > > > > > > > > **3**
> > > > > > > > > > > > > > I am not sure what is the undiscounted state-action visitation since if you sum up to infinite (for infinite horizon case), the visitation would be infinite unless you divide by the time step (e.g., the average visitation in [1]).
> > > > > > > > > > > > >
> > > > > > > > > > > > > Yes, we divide by the time step. Since “undiscounted” is the antonym of “discounted” and since [1] (and many other OPE papers) refers the case $\gamma \in (0,1)$ as “discounted case”, “undiscounted” should naturally refers to the case $\gamma = 1$.
> > > > > > > > > > > > >
> > > > > > > > > > > > > **4**
> > > > > > > > > > > > > > I don’t see how the probability distribution induced from policy rollouts equals to the stationary distribution. To sample from the stationary distribution, I think you need rollout the policy for a sufficient number of steps (e.g., mixing time) and collect the data afterwards.
> > > > > > > > > > > > >
> > > > > > > > > > > > > To the best of our knowledge, assuming that the offline dataset consists of samples from the stationary distribution is standard in both offline-RL and OPE. For example, [1] says “when $\gamma = 1$…, $d_\pi$ is the stationary distribution of $s_t$ as $t \rightarrow \infty$ under policy $\pi$, … we use $(s,a) \sim d_\pi$ to denote draws from distribution $d_\pi(s,a):=d_\pi(s) \pi(a | s)$.”
> > > > > > > > > > > > >
> > > > > > > > > > > > > **5**
> > > > > > > > > > > > > > I would argue this is one of many policy-regularization approaches and one can also do state-action joint policy regularization (as used in this paper).
> > > > > > > > > > > > >
> > > > > > > > > > > > > We do not find reference for “state-action joint policy regularization” in the offline-RL literature. We appreciate your pointing to us some prior work on this topic.
> > > > > > > > > > > > >
> > > > > > > > > > > > > **6**
> > > > > > > > > > > > > > Essentially, both approaches are regularizing the policy to choose actions closer to the behavior policy.
> > > > > > > > > > > > >
> > > > > > > > > > > > > As discussed in our paper, a successful matching of the state-action visitations will also lead to the current policy choosing actions “close to” the behavior policy, since otherwise, the visitation frequency of the current policy will be different from that of the behavior.
> > > > > > > > > > > > >
> > > > > > > > > > > > >
> > > > > > > > > > > > > [1] Qiang Liu, et al. “Breaking the Curse of Horizon: Infinite-Horizon Off-Policy Estimation.” In NeurIPS 2018.
> > > > > > > > > > > > >
> > > > > > > > > > > > > [2] Sergey Levine, et al. “Offline Reinforcement Learning: Tutorial, Review, and Perspectives on Open Problems.” In arXiv:2005.01643.

---

> ### Author Response · Authors · 2021-11-24
> **Response to Reviewer Ht1b After Updating the Review**
>
> Dear Reviewer Ht1b,
>
> We appreciate your responding to our rebuttal and updating your review. It appears to us that you have been making the assumption that as approximating the state-action visitation of a target policy is still an ongoing topic in the OPE community, our proposed state-action joint regularization framework for offline RL can neither be trusted nor be used to explain its empirical success.
>
> We note that OPE and offline RL are related but distinct problems, where OPE focuses on evaluating a target policy while offline RL focuses on learning a policy. The inability to accurately evaluate a certain objective does not imply the inability to maximize/minimize it via a surrogate objective. For example, the Jensen-Shannon divergence is difficult to estimate well, but we can introduce GAN to approximately minimize it. In the context of our paper, the state visitation frequency of the behavior policy is used as a surrogate of that of the target policy and is plugged into our state-action joint regularization framework to help learn the policy in an offline RL setting.

---

### Official Review · Reviewer_H1Xc · 2021-11-05

**Correctness:** 3
**Technical Novelty And Significance:** 2
**Empirical Novelty And Significance:** 3
**Recommendation:** 6
**Confidence:** 4

**Main Review:**

Strengths:
1. The idea of learning policy with latent/implicit structure can be promising in offline RL as the latent structure might suffer less from the distribution shift. The paper borrows some method from GAN training which is novel to offline RL and show they are effective.
2. The paper proposed a number of practical enhancement that can be generalized to a large range of offline policy learning algorithms in section 4.2. These methods are well motivated and tested in the ablation study. I think this study is helpful for many offline RL work.

Weakness:
1. The theory in section 3 is weak and closely related to many previous results. In Kumar et al. (2019), Theorem 4.2 also showed a bound on the visitation ratio when the policy distance is small enough. The results is very similar to this in some sense. However both of them are not really enough to support the claim that it is sufficient to require the policy distance to be small to bound the visitation distance. The key part is in k_max. Assuming policy distance smaller than 1/k_max which is probably requiring some exponentially (in horizon) small number, and cannot hold in any meaningful scenarios. In the end, any constant policy difference can still propagate exponentially to the state visitation distance.
2. The key algorithmic contribution, $L_g(\phi)$, seems vague in some sense. Since the state distribution is fixed, and the state samples in x and y are two i.i.d. samples, the loss can be decoupled into two parts: Construct another example $x' = (\tilde{s}, a')$ where $\tilde{s}$ is the same as in $x$ and $a' \sim \pi_b$. The the distance between $x'$ and $y$ is a constant, then the regularization loss is actually optimizing the conditional action distribution between $\pi_\phi$ and $\pi_b$, which seems not so different from previous work, and it is not so clear why the form in paper is more promising.

**Summary Of The Paper:**

This paper proposed a regularized policy learning algorithm for offline reinforcement learning. The implicit policy is trained by a GAN-like framework, and the regularization loss constrains the distance between learned policy and behavior policy. Experiments and ablation study on the D4RL dataset validate the proposed framework and algorithmic designs.

**Summary Of The Review:**

Overall, this paper proposed a novel algorithmic idea in offline RL. I think the algorithmic contribution in this paper can be helpful to the community. Though some theoretical claim/results in the paper needs to be better explained.

---

> ### Author Response · Authors · 2021-11-15
> **Response to Reviewer H1Xc (Part 1)**
>
> **1.1**
> > The theory in section 3 is weak and closely related to many previous results. In Kumar et al. (2019), Theorem 4.2 also showed a bound on the visitation ratio when the policy distance is small enough. The results are very similar to this in some sense.
>
> Theorem 4.2 in Kumar et al. (2019) [1] states that the concentrability coefficient $C(\Pi_\epsilon)$ of the policy class $\Pi_\epsilon$ is bounded, where $\Pi_\epsilon$ is the set of policies that have support in the probable regions of the behavior policy. To the best of our understanding, the purpose of Theorem 4.2 in Kumar et al. (2019) is to allow a bound on the performance of approximate distribution-constrained Q-iteration (Theorem 4.1) when the set of policies used in the maximization of the Q-function is the policy class $\Pi_\epsilon$. The object of their analysis is *policy class*. More generally speaking, the theoretical analysis in Section 4.1 of Kumar et al. (2019) analyzes the tradeoff between keeping policies chosen during backups close to the data and keeping the constraint set $\Pi$ large enough to capture well-performing policies.
>
> By contrast, the purpose of our theoretical analysis is to show that the prior work in offline RL of controlling the distance between the behavior policy and current policy, in essence, controls their corresponding state-action visitations. The object of our analysis is *policy* and our analysis is conducted on a general setting without specific requirements on the underlying MDP, *e.g.*, a discounted MDP ($\gamma < 1$). To the best of our knowledge, such analysis has not been clearly done in the offline RL literature. Furthermore, this analysis theoretically links a line of prior offline-RL work with our proposed framework, manifesting the generality of our viewpoint and proposed framework.
>
> Therefore, our theoretical analysis is different from the theory in Kumar et al. (2019) in terms of the analysing objects and the purposes of analysis. We agree with the reviewer that the theory in Section 3 could be further strengthened, but would like to note that a thorough analysis of the prior policy-matching offline-RL algorithms is out of the scope of this paper. We leave this for future work.
>
> [1] Aviral Kumar, et al. “Stabilizing Off-Policy Q-Learning via Bootstrapping Error Reduction.” In NeurIPS 2019.
>
> **1.2**
> > However both of them are not really enough to support the claim that it is sufficient to require the policy distance to be small to bound the visitation distance. The key part is in $\kappa_{max}$. Assuming policy distance smaller than $1 / \kappa_{max}$ which is probably requiring some exponentially (in horizon) small number, and cannot hold in any meaningful scenarios. In the end, any constant policy difference can still propagate exponentially to the state visitation distance.
>
> We do not find in our proof the necessity of assuming the policy distance or the $1 / \kappa_{max}$ constant to be some exponentially (in horizon) small numbers. The quantity $1 / \kappa_{max}$ depends on the behavior policy $\pi_b$ and the environmental dynamic $P(\cdot  | s, a)$, which is a constant for a given behavior policy under a specified environment (MDP).
>
> Our theoretical analysis shows that when the policy distance is small, the corresponding visitation distance is also small, where the distance between policies is measured by the largest deviation in the resulting state-conditional action distributions across the state space, similar to the analysis in [1]. The goal of our theoretical analysis, as discussed in Response **1.1**, is to draw a theoretical link between the prior line of research in offline policy-regularization algorithms with our algorithmic idea and proposed framework. Recall that the purpose of these policy-regularization algorithms is to regularize the learned policy towards the behavior policy at all possible states, so as to avoid the detrimental effect of out-of-distribution state-action pairs. Since the effectiveness of these algorithms has been demonstrated, one may reasonably infer that they can achieve their goal of regularizing the state-conditional action distributions to a small distance. Hence our assumption of a small policy distance can be meaningful, especially towards the end of the training process.
>
> When the policy distance is large, it may be hard to analyze the visitation distance without assumptions on the underlying MDP, which we will leave for future work.
>
> [1] John Schulman, et al. “Trust region policy optimization.” In ICML 2015.

---

> > ### Author Response · Authors · 2021-11-15
> > **Response to Reviewer H1Xc (Part 2)**
> >
> > **2.1**
> > > The key algorithmic contribution, $L_g(\phi)$ seems vague in some sense. Since the state distribution is fixed, and the state samples in $x$ and $y$ are two i.i.d. samples, the loss can be decoupled into two parts: Construct another example $x’ = (\tilde{s}, a’)$ where $\tilde{s}$ is the same as in $x$ and $a’ \sim \pi_b$. The distance between $x’$  and $y$ is a constant, then the regularization loss is actually optimizing the conditional action distribution between $\pi_\phi$ and $\pi_b$, which seems not so different from previous work …
> >
> > The regularization term,  $L_g(\phi)$, in our instantiation of approximately minimizing JSD via GAN, is defined as $L_g(\phi) = \mathbb{E}_x[\log(1−D_w(x))]$, which is the generator loss in the classical GAN structure [1].
> >
> > In regularizing the conditional action distribution between $\pi_\phi$ and $\pi_b$, prior works use *the same state* in both the generator sample $x$ and the data sample $y$, as discussed in [2] and [3]. In our work, as shown in Equation 6 (page 6) and Equation 9 (page 7), we construct the generator sample $x$ by making a new sample of state independent of the state in $y$ and adding noise to it, which is different from prior works on policy regularization.
> >
> > As discussed in Section 4.1, directly sampling from the state visitation of the current policy $d_\phi(s)$ may not be feasible in offline RL and as per the comment of Reviewer Ht1b, approximation strategies to it is still an open problem in the OPE community. In our work, we are motivated by the offline policy optimization and prior works in offline RL to use the state visitation of the behavior policy, $d_b(s)$, to approximate that of the current policy, $d_\phi(s)$. Combined with the state-smoothing technique we develop in Section 4.2, our approximation is akin to a kernel density approximation to $d_\phi(s)$ with data points $s \in D$ and with a radial basis kernel of bandwidth $\sigma_J$. Based on the experimental results, our approximation may still effectively regularize the state-action visitation of the current policy to be close to that of the behavior policy during the training process.
> >
> > [1] Ian J. Goodfellow, et al. “Generative Adversarial Nets.” In NeurIPS 2014.
> >
> > [2] Aviral Kumar, et al. “Stabilizing Off-Policy Q-Learning via Bootstrapping Error Reduction.” In NeurIPS 2019.
> >
> > [3] Yifan Wu, et al. “Behavior Regularized Offline Reinforcement Learning.” In arXiv:1911.11361.
> >
> > **2.2**
> > > … and it is not so clear why the form in paper is more promising.
> >
> > As discussed in Section 4.1, our choice of directly matching the state-action joint visitations circumvents the problem in the policy-regularization approach of fitting each state-conditional action distributions on only one data point, since state-action pairs in the offline dataset can all be viewed as samples from the joint visitation frequency, instead of each pair being separately viewed as one sample from the corresponding state-conditional action distribution, *i.e.*, $a_i \sim \pi(\cdot | s_i)$.
> >
> > Our algorithmic design implicitly encourages the smoothness of the state-action mapping, namely, similar states should have similar actions, since the discriminator in GAN can easily discriminate a generator sample $x$ as “fake” should its state be similar to a data sample but action very different from. This smoothness feature implicitly helps a reliable generalization of our policy to unseen states.
> >
> > Our enhancement technique of state-smoothing, developed in Section 4.2, further provides a better coverage of the state space and explicitly encourages predictable and smooth behavior at states unseen in the offline dataset.

---

### Author Response · Authors · 2021-11-15
**Revision Summary**

We would like to thank the reviewers for their detailed comments. Below we provide a detailed point-by-point response. We have also updated the paper to address the reviewers’ suggestions and concerns and highlighted the major modifications in blue color. Summary of the updates are as follows:

1. We refine the description of the purpose of our theoretical analysis in Section 3;
2. We modify the statement of Theorem 1 and Theorem 2 in Section 3 to clearly show that we bound the total variation distance between the state visitation frequencies;
3. We add in Section 3 a comparison between the bound in this paper and that in the prior work on the discrepancy between state visitation frequencies;
4. We add the evaluation metric Average Rank to Table 1 (page 9), which provides a robust algorithmic evaluation;
5. Due to the space constraint, we move the description of our experimental datasets to Appendix D.2 and add references for our dataset choice.

***
(Update on Nov. 20, 2021)

1. We update the comparison in Section 3 between the bound in this paper and that in the prior work on the discrepancy between state visitation frequencies.

---

### Author Response · Authors · 2021-11-21
**Call for Comments on Further Revising the Paper**

Dear reviewers,

We notice that there are still significant disagreements on the quality of our paper. As the revision deadline is approaching, we appreciate your letting us know if there are any specific parts of the paper that you'd like us to further revise.

---

> ### Comment · Area_Chair_S3cF · 2021-11-22
> **RE:**
>
> Looking at the experimental results, I had a couple of questions:
> 1) How was the subset D4RL tasks selected for Table 1? Why were some omitted?
> 2) There have been a number of papers that have improved upon CQL, including S4RL (Sinha et al.) which part of your method draws inspiration from. I would like to see S4RL (and other relevant state-of-the-art-methods) added as a comparison to Table 1 or an explanation for why the existing baseline algorithms are sufficient.

---

> > ### Author Response · Authors · 2021-11-23
> > **Response to Area Chair S3cF**
> >
> > Dear AC,
> >
> > Thank you very much for your questions. Below are our responses.
> >
> > **1**
> > > How was the subset D4RL tasks selected for Table 1? Why were some omitted?
> >
> > The D4RL benchmark is comprehensive. We are only able to test on a representative subset of the datasets therein given our limited computational resources. Besides, we notice that the D4RL benchmark is still under active development, and new datasets are frequently added. To save computational power, we have focused on datasets that have been benchmarked in the official whitepaper [1].
> >
> > Below we briefly discuss why we omit some domain/datasets:
> >
> > 1. AntMaze: We omit the AntMaze domain of datasets since the reward/return scales in these datasets are different from the Gym-MuJoCo domain. Therefore, based on the CQL source code, these datasets can require non-trivial reward scaling/transformation and hyperparameter settings significantly different from the Gym-MuJoCo domain. Given our limited computational resources, we decided to leave the AntMaze domain for future work. Additionally, the Maze2D datasets we test on have similar nature with the AntMaze datasets in terms of non-Markovian data-collecting policies, but have reward/return scales similar to the Gym-MuJoCo domain.
> > 2. Gym-MuJoCo: As discussed in Appendix D.2 of our paper and in the response to Reviewer Ht1b, we do not test on the “random” and the “expert” datasets as they are less practical [2] and could be respectively solved by directly using standard off-policy RL algorithms [3] and the behavior cloning algorithms.
> > 3. Adroit: Based on the results reported in the D4RL whitepaper [1] and considering our computational budget, we select four datasets that prior algorithms can generally learn meaningful policies, across all three types of the datasets (“human”, “cloned”, and “expert”). We notice that on several datasets of this domain, prior algorithms generally fail. We restrain ourselves from testing on these datasets and would rather wait for more discussions in the community, *e.g.*, on the dataset/data quality.
> > 4. Flow, Franka Kitchen and CARLA: To the best of our knowledge, these domains of datasets have been rarely tested in the offline RL literature, and can be computationally intensive. To save computational resources, we choose to omit these datasets.
> >
> > [1] Justin Fu, et al. “D4RL: Datasets for Deep Data-Driven Reinforcement Learning.” In arXiv:2004.07219.
> >
> > [2] Tatsuya Matsushima, et al. “Deployment-Efficient Reinforcement Learning via Model-Based Offline Optimization” In ICLR 2021.
> >
> > [3] Rishabh Agarwal, et al. “An Optimistic Perspective on Offline Reinforcement Learning”. In ICML 2020.
> >
> > **2**
> > >  I would like to see S4RL (and other relevant state-of-the-art-methods) added as a comparison to Table 1 or an explanation for why the existing baseline algorithms are sufficient.
> >
> > As this is a very active research area, it is very possible that we have missed some very recent baselines that have improved on CQL. We would appreciate it if you could point out specific ones for us to focus on. We would try our best to add these additional comparisons in the next few days, assuming there is an official codebase that is publicly available and the paper appeared sufficiently long before the ICLR submission deadline. Note that we have not compared to the recently published S4RL as it has not yet made its code publicly available and we are not able to reproduce its results using our own implementation.

---

### Decision · Program_Chairs · 2022-01-20

**Decision:**

Reject

**Comment:**

The authors propose to use implicit policies (similar to a conditional GAN) with a GAN-inspired regularizer. Theoretically, they show an equivalence between policy-matching and state-action-visitation matching. Finally, they evaluate their approach on D4RL and showed improved performance as well as ablations.

Reviewers did not find the theoretical contribution to be significant.  While the exact form may be novel, the general result has been shown in previous work and they only use the general result as a loose motivation for their approach. All reviewers acknowledge their empirical improvements as the primary strength of the paper. While a central component of their story is joint state-action regularization, Reviewer Ht1b identified that their proposed approach does not appear to directly regularize the joint state-action distribution, but rather behaves more similarly to existing policy constraint methods. I agree with Reviewer Ht1b and after much back-and-forth discussion (both Reviewer Ht1b and myself) with the authors, I have not been persuaded otherwise.

The paper has a lot of potential - strong empirical results, but the justification and explanation of the method needs to be rewritten in light of the policy constraint regularization or a stronger argument needs to be put forth in support of joint state-action regularization. I don't think this diminishes the results though, but without this substantial revision, I cannot accept the paper at this time.